# C²P: Featuring Large Language Models with Causal Reasoning

## Abstract

Causal reasoning is one of the primary bottlenecks that Large Language Models (LLMs) must overcome to attain human-level intelligence. Recent studies indicate that LLMs display near-random performance on reasoning tasks. To address this, we introduce the Causal Chain of Prompting (C²P), the first reasoning framework that equips current LLMs with causal reasoning capabilities. C²P operates autonomously, without relying on external tools or modules during both the causal learning and reasoning phases, and can be seamlessly integrated into the training or fine-tuning of LLMs. To evaluate the performance of C²P, we first demonstrate that reasoning accuracy improved by over 30.7% and 25.9% for GPT-4 Turbo and LLaMA 3.1, respectively, when using our framework, compared to the same models without C²P on a synthetic benchmark dataset. Then, using few-shot learning of the same LLMs with C²P, reasoning accuracy increased by over 20.05% and 20.89%, respectively, with as few as ten examples, compared to the corresponding LLMs without C²P on the same dataset. To better evaluate C²P in realistic scenarios, we utilized another benchmark dataset containing natural stories across various fields, including healthcare, medicine, economics, education, social sciences, environmental science, and marketing. The results show improved reasoning when C²P is applied, compared to cases where our framework is not used, which often leads to random or hallucinated responses. The improvement observed in both few-shot learned GPT-4 Turbo and LLaMA 3.1 provides evidence of the generalizability of C²P, highlighting its potential to be incorporated into the training or fine-tuning of new LLMs to enhance their reasoning capabilities.

## 1 Introduction

Recent advancements in Large Language Models (LLMs) have impacted existing AI paradigms and raised expectations regarding AI's capabilities (Achiam et al., 2023; Brown, 2020). LLMs generally produce outputs based on the most likely results learned from vast amounts of training data (Vaswani et al., 2017). This enables them to acquire extensive knowledge, ranging from common sense to specialized domains such as mathematics and science (Jiralerspong et al., 2024). Numerous examples of interventions, outcomes, and explanations are included in their training of LLMs to reduce hallucinations and improve reasoning. However, despite significant architectural differences, hallucinatory responses still occur and true causal reasoning remains lacking (Kalai & Vempala, 2023; Xu et al., 2024). As a result, although models may appear to reason causally, they do not engage in a genuine causal reasoning process (Zečević et al., 2023). This deficiency represents a fundamental limitation of LLMs as AI systems compared to human intelligence, which is based on causal reasoning rather than simple associations for decision making (Penn & Povinelli, 2007; Anwar et al., 2024). Judea Pearl introduced "The Ladder of Causation" in (Pearl & Mackenzie, 2018) that easily addresses the reason for this deficiency. This ladder includes three main levels. At the first level, association, only patterns and dependencies are identified, and why things are related cannot be answered. The second level, intervention, involves understanding cause-and-effect by predicting outcomes of actions or changes, asking "What happens if I do this?" The highest level, counterfactuals, deals with imagining alternative scenarios, asking "What would have happened if...?". This framework illustrates how

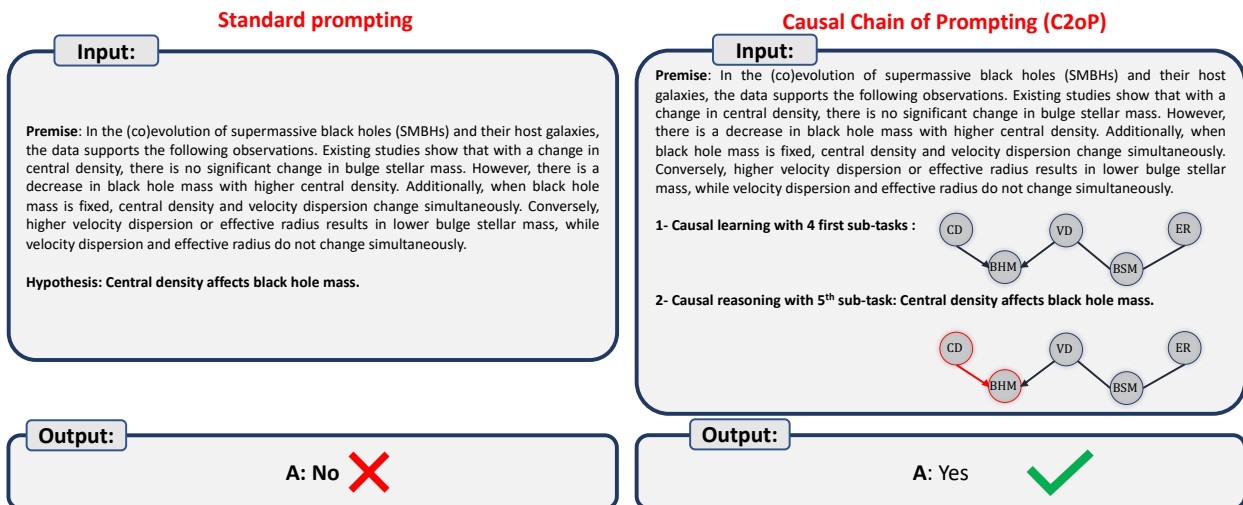

Figure 1: Example of the standard prompting vs few-shot learned GPT-4 with $C^2P$ in open problem in astrophysics (Pasquato et al., 2023)

.

reasoning evolves from pattern recognition to understanding and predicting causal relationships, highlighting the differences at each level.

Recently, studies on the causal reasoning capabilities of LLMs have garnered significant interest, with most focusing on evaluating the models' reasoning ability and a few on improving this feature. The inefficiency of LLMs in reasoning has been extensively studied from various perspectives, as demonstrated in works such as (Xu et al., 2023; Romanou et al., 2023; Jin et al., 2023a). Similarly, Jin et al. (2023b) introduces the CORR2CAUSE dataset, revealing that current models often perform no better than random chance when answering causal questions. Most recently, simple tasks have been shown to completely break down the reasoning abilities of state-of-the-art LLMs (Nezhurina et al., 2024). Additionally, studies such as (Petroni et al., 2019; Jiang et al., 2020) aimed to reason causally based on the knowledge already present in the training data of LLMs, which is why LLMs are referred to as "causal parrots" in (Zečević et al., 2023). The reason for these behaviors is discussed in (Wang et al., 2023; Imani et al., 2023). As one of the initial attempts to enhance reasoning in LLMs, chain-of-thought prompting is presented in (Wei et al., 2022b), showing improvement based on the data from the given query. As another approach, LLMs have been utilized in conjunction with external tools to extract causal structures, as demonstrated in (Jiralerspong et al., 2024). More recently, in (Ashwani et al., 2024), a novel architecture called the Context-Aware Reasoning Enhancement with Counterfactual Analysis (CARE-CA) framework is presented to enhance causal reasoning and explainability. Their proposed framework incorporates an external explicit causal detection module with ConceptNet (Speer et al., 2017) and counterfactual statements, as well as implicit causal detection through LLMs, showing progress in causal reasoning in short and simple queries. Several other works at the intersection of causal inference and LLMs are discussed in an extensive survey by Liu et al. (2024). The main drawbacks of the existing frameworks aiming to equip LLMs with reasoning are their reliance on external modules, the need for extensive information to function, and their very low accuracy.

In this paper, we propose the first reasoning framework for LLMs, called the Causal Chain of Prompting ($C^2P$), designed to enhance reasoning skills by climbing the causality ladder to address reasoning questions. Unlike existing methods, $C^2P$ operates autonomously, without relying on external tools or modules during the learning and reasoning phases. It can be easily implemented in the few-shot, fine-tuning, or training process of LLMs to improve reasoning in causal questions. Importantly, $C^2P$ extracts the causal relation based on associations mentioned in the given premise. $C^2P$ is inspired by Pearl's foundational work, which argues that causal Directed Acyclic Graphs (DAGs), along with d-separation, enable the investigation of cause-and-effect relationships without relying on structural equation models in computational studies (Pearl,

1995). Based on this, we demonstrate that by identifying the adjacency matrix of causal relationships among variables in the premise—similar to, but distinct from, the causal DAG in Pearl's framework—a reasoning query can be effectively answered. This framework includes five simple main subtasks, as follows: (1)Prompting to extract random variables from the provided data. (2)Prompting to extract all conditional and unconditional associations, as well as cause-and-effect relations specifically mentioned among the random variables. (3)Prompting to create the initial adjacency matrix with values of 1 for all elements except the diagonal elements and those corresponding to effect-cause relations (the cause-and-effect elements are also set to 1). (4) Prompting conditional and unconditional independencies and identification of colliders, step by step, to extract the causal adjacency matrix. (5)Prompting for reasoning questions or hypotheses. (see Fig. 1). To evaluate the accuracy and reliability of implementing the $C^2P$ on LLMs, we initially assess it using publicly available benchmark synthetic and "Natural Story" datasets, such as those in (Jin et al., 2023b). To demonstrate the practical applicability of $C^2P$, we present the results of few-shot learned LLMs with $C^2P$ in both synthetic and real-world scenarios. Subsequently, we evaluate it in more realistic and complex scenarios found in real-world problems presented in (Pasquato et al., 2023).

**Contributions.** In this work, we present several important contributions to facilitate causal reasoning in language models. Concretely,

1. We introduce the $C^2P$ framework as the first reasoning framework to equip LLMs with causal reasoning within real-world scenarios, without relying on external tools.

2. Through extensive experiments with our framework, we demonstrate a significant improvement of LLMs in causal reasoning in various benchmarks. Additionally, we examine the performance of the $C^2P$ framework on more complex and real-world scenarios in various domains.

3. By implementing $C^2P$ in the few-shot learning process of both GPT-4 Turbo and LlaMA 3.1, and demonstrating improvements in reasoning, we highlight the evidence for the generality of our approach.

The codes for sections 3 and 4 are publicly available at `https://anonymous.4open.science/r/C2P-5C2A`.

## 2 Preliminaries on Causal Learning and Reasoning

To lay the groundwork for our framework, understanding the fundamental concept of cause and effect is paramount. A widely acknowledged principle in causality is the principle proposed by Reichenbach, which posits the following.

**Common cause principle (Reichenbach, 1991)** If two random variables $X_1$ and $X_2$ are statistically dependent, i.e., $X_1 \not\perp X_2$, then there exists a third variable $X_3$ that causally influences both. (As a special case, $X_3$ may coincide with either $X_1$ or $X_2$.) Furthermore, this variable $X_3$ screens $X_1$ and $X_2$ from each other in the sense that given $X_3$, they become independent, $X_1 \perp\!\!\!\perp X_2 | X_3$.

**do calculus (Pearl, 1995):** do-calculus, developed by Judea Pearl et al., is a set of rules used to transform and manipulate causal expressions within causal diagrams (or graphical models). do-calculus is a formal tool used to reason in causal relationships from a mixture of experimental and observational data. do-calculus consists of three main rules that allow one to rewrite expressions involving interventions (typically represented as $do(x)$, indicating an intervention to set variable $X$ to value $x$). These rules are crucial for determining the identifiability of causal effects from data, allowing researchers to reason about causal relationships using a combination of experimental and observational data. The rules are provided in Appendix A.1.

**d-separation (Pearl, 1995).** d-Separation is a criterion used in Bayesian network analysis to determine whether a set of variables $X_1$ is independent of another set of variables $X_2$, given a third set of variables $X_3$. This concept is foundational in understanding the flow of causal effects in graphical models and helps in deciding whether a path between two variables is "blocked" or not by conditioning on other variables. According to the d-separation, a path between two variables is blocked if it includes an intermediate variable that is a collider and is not conditioned on or a non-collider that is conditioned on. Here, a **collider** is a

variable that has arrows inward from two other nodes (i.e., $X_1 \rightarrow X_2 \leftarrow X_3$), whereas a **non-collider** does not meet this criterion.

**Directed Acyclic Graphs.** A graph $G$ is called a Partially Directed Acyclic Graph (PDAG) if there is no directed cycle, that is, if there is no pair $(X_j, X_k)$ with directed paths from $X_j$ to $X_k$ and from k to $j$. $G$ is called a Directed Acyclic Graph (DAG) if it is a PDAG and all edges are directed.

**Markov Property.** The Markov property in a DAG $G$ states that each node $X_i$ is conditionally independent of its non-descendants, given its parents. In other words, $X_i \perp\!\!\!\perp \text{NonDe}(X_i)|\text{Pa}(X_i)$, where $\text{NonDe}(X_i)$ represents the non-descendants of $X_i$, excluding itself, and $\text{Pa}(X_i)$ represents the parents of $X_i$. It helps factorize the distribution of all graph nodes as $P(X_1, \ldots, X_N) = \prod_{i=1}^{N} P(X_i|\text{Pa}(X_i))$.

**Faithfulness.** This assumption ensures that all the d-separation sets in the graph can be inferred from the independence relations in the distribution. In the sequel, we assume faithfulness, a widely used assumption in causal discovery (Spirtes et al., 2001).

**Markov Equivalence of Graphs.** Two DAGs are Markov equivalent if they generate the same joint distribution, $P(\mathbf{X})$. A set of DAGs that are Markov equivalent is referred to as a Markov equivalence class (MEC). Causal graphs within the same MEC are easily recognizable because they share the same skeleton (i.e., the same undirected edges) and the same V-structures (i.e., configurations like $X_1 \rightarrow X_2 \leftarrow X_3$, where $X_1$ and $X_3$ are not directly connected).

**Ladder of Causation (Pearl & Mackenzie, 2018).** To perform any level of causal reasoning, the Ladder of Causation proposes three main levels: "seeing or association," "doing or intervention," and "imagining or counterfactual". The questions that can be answered in Association (Seeing) are mostly similar to "What is happening?" This is the most basic level where we observe patterns and correlations between variables. At this level, we can only say that two things are related or tend to occur together, but we cannot explain why. Machine learning models and statistical methods that rely on pattern recognition (like LLMs) often operate at this level. In the second level, Intervention (Doing), questions such as "What happens if I do something?" can be addressed. At this level, we can go beyond mere association and ask about the effect of an action or intervention. This requires understanding the cause-and-effect relationships. To reach the third level of the ladder, more information is needed on the causal structure, which can mainly be provided with structural causal models (SCMs), assuming that all assumptions are satisfied (Bareinboim et al., 2022). However, studies such as (Spirtes et al., 2001) have shown that Level 2 can be reached (up to equivalence classes) using only observational data.

Note that the current causal discovery methods are primarily divided into two groups: (i) Constraint-based algorithms such as the PC algorithm (Spirtes et al., 2001), which has a PDAG as an output, which represents the MEC of the true underlying graph and is the best outcome that these methods can achieve; (ii) Score-based methods such as GES (Chickering, 2002), NOTEARS (Zheng et al., 2018), GOLEM (Ng et al., 2020), DAGMA (Bello et al., 2022), and TOPO (Deng et al., 2023), among many others, which extract a DAG that mostly fits the data and demonstrated high accuracy in extracting Bayesian networks. Nevertheless, score-based methods require the solution of a numerical optimization problem, making them difficult to integrate with LLMs.

**The PC algorithm (Spirtes et al., 2001).** The PC algorithm is developed based on Reichenbach's common cause principle and the Markov property, its steps can be described as follows:

(i) Form a complete undirected graph

(ii) Eliminate edges between variables that are unconditionally independent

(iii) For each pair of variables $(X_1, X_2)$ having an edge between them, and for each variable, $X_3$ with an edge connected to either of them, eliminate the edge between $X_1$ and $X_2$ if $X_1 \perp\!\!\!\perp X_2|X_3$

(iv) For each pair of variables $X_1, X_2$ having an edge between them, and for each pair of variables $\{X_3, X_4\}$ with edges both connected to $X_1$ or both connected to $X_2$, eliminate the edge between $X_1$ and $X_2$ if $X_1 \perp\!\!\!\perp X_2|\{X_3, X_4\}$.

(v) For each triple of variables $(X_1, X_2, X_3)$ such that $X_1$ and $X_2$ are adjacent, $X_2$ and $X_3$ are adjacent, and $X_1$ and $X_3$ are not adjacent, orient the edges $X_1 - X_2 - X_3$ as $X_1 \rightarrow X_2 \leftarrow X_3$, if $X_2$ was not in the set conditioning on which $X_1$ and $X_3$ became independent and the edge between them was accordingly eliminated. We call such a triple of variables a V-structure.

(vi) For each triple of variables such that $X_1 \rightarrow X_2 - X_3$, and $X_1$ and $X_2$ are not adjacent, orient the edge $X_2 - X_3$ as $X_2 \rightarrow X_3$. This is called orientation propagation.

## 3  Developing Causal Chain of Prompting

To develop C$^2$P framework, we aim to extract the adjacency matrix of variables in a given premise. The adjacency matrix acts as equivalent to and instead of the causal DAG in Pearl's work, leading to answering the causal reasoning question. In developing the Causal Chain of Prompting, we use a few steps similar to the PC algorithm for the given premise.

### 3.1  Causal Chain of Prompting (C$^2$P)

The C$^2$P framework consists of five main subtasks as follows, for learning and reasoning about cause and effect relations for the given premise:

- **Subtask 1:** Prompting to extract the random variables in the provided data.

- **Subtask 2:** Prompting to extract all the cause-and-effect relations along with all conditional and unconditional relations among the random variables specifically mentioned in the given premise.

- **Subtask 3:** Prompting to create an initial adjacency matrix where all elements are 1, except for the diagonal elements and those corresponding to the cause-and-effect relationships specifically mentioned in the given premise (extracted in subtask 2).

- **Subtask 4:** Prompting of conditional and unconditional independence evaluation and identification of colliders to extract the causal PDAG.

- **Subtask 5:** Prompting for cause-and-effect questions or hypotheses.

Each subtask in C$^2$P can include one or multiple steps (prompts). In general, to execute the framework, 9 main steps must be completed. **Subtask 1** is completed in Step 1. **Subtask 2** is accomplished with Step 2. **Subtask 3** is achieved with Step 3. To perform **Subtask 4**, 5 steps must be applied. Step 4 first eliminates all unconditional independencies achieved in Subtask 2 and Step 5 then removes all conditional independencies extracted in Subtask 2. Step 6 identifies potential nodes that can act as colliders. Step 7 confirms whether the nodes identified in Step 6 are colliders. Step 8 updates the adjacency matrix, resulting in the final adjacency matrix of a causal structure. To perform **Subtask 5**, the causal question is asked in Step 9.

Fig. 2 illustrates the five subtasks of C$^2$P for applying the PC algorithm. The exact prompts for all the steps are provided in Appendix A.3.

### 3.2  Few-shot learning

LLMs have powerful zero-shot capabilities, yet they struggle with complex tasks because of their engineering designs and insufficient examples in the training process. In such situations, few-shot learning is a versatile and efficient technique for in-context learning, which can be used to quickly adapt LLMs to new tasks and significantly enhance their performance (Min et al., 2022; Touvron et al., 2023). This approach involves providing several examples with desired answers to condition the LLMs to produce correct responses for new instances with similar patterns. The few-shot learning process of C$^2$P, as described in the previous subsection, is based on more abstract prompts (due to token limitations). The prompts and an example of the given story are included in Appendix A.4. Depending on the token limitations of the employed LLMs,

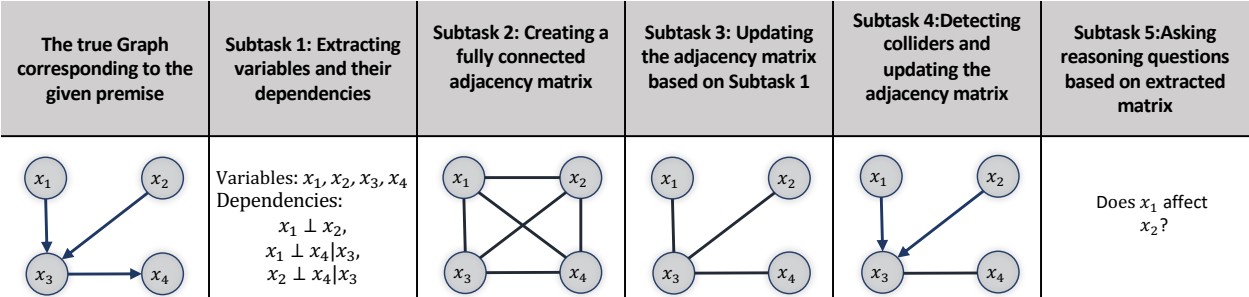

Figure 2: Example of applying the five subtasks of $C^2P$

.

the number of examples (shots) may vary. For GPT-4 Turbo, which has a 30,000 token limit, we were able to include ten examples. For LLaMA 3.1, we used the same examples in the few-shot learning process as well.

It is important to note that steps from other constraint-based methods can either replace or complement the steps of the PC algorithm in $C^2P$ methods, as comprehensively discussed in (Glymour et al., 2019). For instance, the steps in the FCI method can be used in cases where the causal sufficiency assumption is violated, i.e., where latent variables and selection bias may be present (Spirtes et al., 1995). Additionally, the classifiers proposed by Ceraolo et al. (2024) formalize the definition of causal questions and establish a taxonomy for finer-grained classification. These classifiers can be used before our framework to identify the causal question as a prerequisite of our method.

## 4 Results of Experiments for Featuring LLMs with Causal Reasoning

### 4.1 Datasets

The experiments are divided into two data groups: a synthetic dataset, a realistic scenarios dataset, and a real-world example.

**The CORR2CAUSE dataset:** The CORR2CAUSE dataset introduced in (Jin et al., 2023b), serves as a benchmark to assess the ability of LLMs to respond to reasoning queries. The process of creating CORR2CAUSE is as follows: First, select the number $N$ of variables (Step 1) and generate all unique DAGs with $N$ nodes (Step 2). Next, gather all d-separation sets from these graphs to identify MECs (Step 3). In Step 4, formalize the data by associating each MEC with its corresponding causal graphs. For each MEC, construct a correlation statement based on the statistical relations within the MEC, hypothesize a causal relationship between two variables, and assign a validity $v = 1$ if the hypothesis holds for all causal graphs in the MEC, or $v = 0$ if the hypothesis does not necessarily apply to all graphs in the MEC. Finally, introduce the verbalization process. An example of a premise and its corresponding hypothesis in the CORR2CAUSE dataset is as follows:

**Premise:** *Suppose that there is a closed system of 3 variables, A, B and C. All statistical relations among these 3 variables are as follows: A correlates with C. B correlates with C. However, A is independent of B.*
**Hypothesis:** *A directly affects C.*

**The Natural Stories dataset:** The Natural Stories dataset is also introduced in (Jin et al., 2023b) to assess the reasoning capabilities of LLMs in realistic scenarios. This dataset builds upon the CORR2CAUSE dataset as a foundation for future extensions in various contexts, such as instantiating variables with real-world phenomena and placing the narratives in more natural settings. For instance, the rule "correlation does not imply causation" can be illustrated using ice cream sales and swimming pool attendance as variables, by arguing that ice cream sales do not necessarily impact swimming pool attendance, as both could be influenced by a third factor, such as hot weather. The Natural Stories data presented in (Jin et al., 2023b)

is not open-source. However, generating such data is straightforward. By providing an example from the CORR2CAUSE dataset along with the corresponding Natural Stories created by a human or existing example on the web, GPT-4 can generate realistic, everyday narratives. For more detailed examples of generating such stories, refer to the code directory accompanying this paper. A natural story generated by GPT-4, inspired by the previous example, could be:

**Premise:** *Let's consider three factors: eating junk food, obesity, and watching television. There is a correlation between eating junk food and obesity, and between watching television and obesity. However, eating junk food and watching television are independent from each other.*

**Hypothesis:** *Eating junk food directly affects obesity.*

More details and examples are provided in the code repository of this study.

**An example on supermassive black holes.** The data and results on the coevolution of supermassive black holes (SMBHs) and their host galaxies, presented in (Pasquato et al., 2023), are used as a real-world example in its verbalized form. The verbalized information is as follows:

**Premise:** *In the (co)evolution of supermassive black holes (SMBHs) and their host galaxies, the data supports the following observations. Existing studies show that with a change in central density, there is no significant change in bulge stellar mass. However, there is a decrease in black hole mass with higher central density. Additionally, when black hole mass is fixed, central density and velocity dispersion change simultaneously. Conversely, higher velocity dispersion or effective radius results in lower bulge stellar mass, while velocity dispersion and effective radius do not change simultaneously.*

**Hypothesis:** *Does central density affect black hole mass?*

### 4.2 Experimental setup

To test existing LLMs on the first synthetic data, we first include two BERT-based NLI models in the transformers library (Wolf et al., 2020): BART (Lewis et al., 2019), DistilBART (Shleifer & Rush, 2020). We evaluate LLaMA3-8B (Touvron et al., 2023) and LLaMA3-70B (Taori et al., 2023). We also evaluate the latest, more efficient models, LLaMA 3.1 8B and LLaMA 3.1 70B released in late September 2024. We assess the GPT-3.5 (i.e., ChatGPT), and the GPT-4 (Achiam et al., 2023), using the OpenAI API (https://openai.com/api/). We also used the latest GPT-4o model, released on October 2, 2024. Then, we evaluate the reliability of $C^2P$ on various queries, each with a different number of random variables. Due to the limitation of tokens for different versions of GPTs, we employed GP4-Turbo which has a 30000 maximum token limit. We also used $C^2P$ with LLaMA 3.1 70B as well. In all of the models, we set the temperature to 0. Note that various types of LLMs were tested in (Jin et al., 2023b), and the results showed that existing LLMs perform worse than random, completely random, or only slightly better than random in responding to such queries. From the models tested in (Jin et al., 2023b), we selected those with the highest accuracies to avoid duplicating the same results and to stay focused on the main objective of this study.

We selected 120 samples at random from the CORR2CAUSE dataset for our experiment, ensuring a balanced distribution with 60 "Yes" and 60 "No" answers. In the CORR2CAUSE dataset, the majority of responses are "No", as a result, they are only able to use the F1 score as the primary metric for accuracy. Our experimental design, which balances "Yes" and "No" responses, enhances the realism and comprehensibility of the computational metrics, providing a more accurate reflection of the model's performance. As a result, all four key metrics—F1 score, precision, recall, and accuracy—are valuable, each offering a unique perspective on the model's capabilities. These samples explore reasoning within three causal scenarios: direct cause-and-effect relationships (Fig. 3 i), indirect cause-and-effect relationships (Fig. 3 ii), and the presence of an effect due to two causes (Fig. 3 iii).

To evaluate the proposed framework for natural stories, we used GPT-4, which excels at story generation. We crafted instructions in the prompts and generated 30 stories for our case study in fields such as healthcare and medicine, economics, social sciences, environmental science, and marketing, all highlighting the importance of causality. This process is similar to the one presented in (Jin et al., 2023b). Our approach can be tested using the examples of Simpson's paradox discussed in (Pearl & Mackenzie, 2018). However, since these

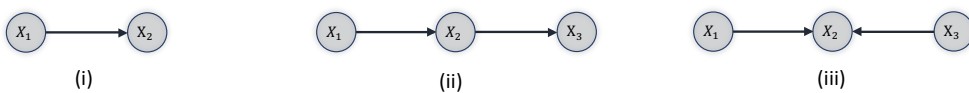

Figure 3: All possible in cause and effect relations. i. $X_1$ directly causes $X_2$ ($X_2$ is directly effect of $X_1$). ii. $X_1$ indirectly causes $X_3$ ($X_3$ is indirectly effect of $X_1$). iii. $X_1$ and $X_3$ are causes of $X_2$ ($X_2$ is common effect of $X_1$ and $X_3$)

examples are already included in the training data of current LLMs, the models simply repeat the correct answers based on that training data, similar to the parrot study in (Zečević et al., 2023). Consequently, our generated natural stories replicate these examples in a manner that the LLM cannot address within its existing training data. We aimed to demonstrate how the symbolic expressions in the CORR2CAUSE dataset can affect the reasoning of our proposed framework.

Finally, we assess the coevolution of supermassive black holes (SMBHs) and their host galaxies using our proposed framework, replicating the results from (Pasquato et al., 2023). The goal in (Pasquato et al., 2023) was to extract a PDAG of SMBHs based on numerical data and then infer causal reasoning questions from the graph. We use verbalized information on SMBHs to evaluate whether the LLM, enhanced with $C^2P$, can answer reasoning questions such as, "Does central density affect black hole mass?"

It is important to note that Chain-of-Thought (CoT) prompting is implemented in most existing LLMs, enhancing their reasoning accuracy, as discussed in (Chung et al., 2024). Consequently, in our experiments, LLMs are prompted to think step by step when responding to reasoning questions, ensuring that the CoT mechanism is activated.

## 4.3 Evaluation of the $C^2P$ on synthetic dataset

**Results of the $C^2P$ on CORR2CAUSE dataset:** In Table 1, we show the performance of LLMs in the cause-effect task with and without employment of $C^2P$ framework.

According to Table 1, performing causal reasoning tasks remains a significant challenge for existing LLMs, even when prompted to think step by step, similar to (Wei et al., 2022b). These results show that even more updated models can sometimes perform worse than older ones in some metrics, for instance, LlaMa 3.1 acts worse than LlaMa 3. Based on the responses presented in the code repository for this study, model hallucination is one of the major factors contributing to this poor performance, aligning with the findings in Jin et al. (2023b). Table 1 shows that by applying 5 consecutive tasks of $C^2P$ on GPT-4 Turbo and LlaMa 3.1, the reasoning accuracy improved by over 30.7% and 25.9%, respectively, compared to the corresponding LLMs without $C^2P$ on a synthetic benchmark dataset. Then, using few-shot learning of GPT-4 Turbo and LLaMA 3.1 with $C^2P$, reasoning accuracy increased by over 20.05% and 20.89%, respectively, with as few as ten examples, compared to the corresponding LLMs without $C^2P$ on the same dataset. To ensure that the improvements are significant in both step-by-step prompting and few-shot experiments, we applied the sample size formula for comparing two proportions from (Chow et al., 2017). This formula indicates that even with a sample size of 117, the observed differences are statistically significant we assume a confidence level of 99% and 80% power. It is important to note that the primary factor contributing to the discrepancy between the results of LLaMA and GPT, and those reported in (Jin et al., 2023b), is the distribution of the provided premise. This table shows that when reasoning questions are posed to GPT models, their responses tend to slightly favor the "No" answer. Interestingly, LLaMA models slightly tend to answer "Yes" to reasoning questions more frequently. It is important to note that even when LLMs provide the correct final answer, the reasoning process that leads to that answer can still be incorrect, as illustrated in A.2, Table 5, for the results of GPT-4. This highlights the randomness of their responses more clearly. Other prompts that can execute the algorithm can be used in place of ours, as long as they perform the same subtasks. This supports the idea presented in the Lu et al. (2024). Furthermore, this becomes particularly evident in the few-shot learning process, where providing more examples has a significant impact; even

Table 1: Comparison of applying $C^2P$ frameworks in LLMs compared to the existing LLMs with CoT

| Models | F1 | Precision | Recall | Accuracy |
|---|---|---|---|---|
| **Random Baselines** | | | | |
| Random (Proportional) | 13.5 | 12.53 | 14.62 | 71.46 |
| Random (Uniform) | 20.38 | 15.11 | 31.29 | 62.78 |
| **BERT-Based Models** | | | | |
| DistilBART MNLI | 26.74 | 15.92 | 83.63 | 30.23 |
| BART MNLI | 33.38 | 31.59 | 35.38 | 78.50 |
| **LLaMA-Based Models** | | | | |
| LLaMA 3-8B | 43.37 | 48.6 | 43.15 | 47.5 |
| LLaMA 3-70B | 49.41 | 53.84 | 62.23 | 55.77 |
| LLaMA 3.1-8B | 53.96 | 45.94 | 61.41 | 46.66 |
| LLaMA 3.1-70B | 57.14 | 48.64 | 49.15 | 48.07 |
| **$C^2P$ with LLaMA 3.1-70B** | **81.63** | **83.3** | **83.3** | **81.66** |
| **$C^2P$ Few-shot learned LLaMA 3.1-70B** | **76** | **79.1** | **73.07** | **76.66** |
| **GPT-Based Models** | | | | |
| GPT-3.5 | 47.5 | 56.2 | 43.2 | 52.5 |
| GPT-4 Turbo | 51.9 | 51.92 | 45 | 54.2 |
| GPT-4 | 50.2 | 54.1 | 47.1 | 55 |
| GPT-4O | 59.86 | 61.46 | 58.33 | 60.83 |
| **$C^2P$ with GPT-4 Turbo** | **91.72** | **93.47** | **89.2** | **91.66** |
| **$C^2P$ Few-shot learned GPT-4 Turbo** | **82.16** | **80.71** | **77.41** | **80.88** |

more abstract prompts can achieve the same or even better results.

**Robustness analysis:** To assess the robustness of the proposed prompts for each of the 5 subtasks, Table 2 shows the computed accuracy computed in each subtask of implementing $C^2P$ (subsection 3.1) on GPT-4 Turbo for different numbers of variables in the given premise.

Table 2: Accuracy by number of variables and subtasks

| Number of variables | Accuracy | | | | |
|---|---|---|---|---|---|
| | First subtask | Second subtask | Third subtask | Fourth subtask | Fifth subtask |
| 3 variables | 100% | 100% | 100% | 99.12% | 98.7% |
| 4 variables | 100% | 100% | 100% | 97.5% | 84.1% |
| 5 variables | 100% | 100% | 100% | 87.5% | 75% |
| 6 variables | 100% | 100% | 100% | 78.3% | 70% |

## 4.4 Evaluation of the $C^2P$ on natural stories

The results of applying the $C^2P$ framework, both step-by-step and few-shot learned for GPT-4 Turbo and LLaMA 3.1, are provided in Table 3 and compared to the results of GPTs and LLaMA 3.1. This table, along with the highlighted sections, clearly demonstrates that the existing methods respond randomly to reasoning prompts. In contrast, GPT-4 Turbo and LLaMA 3.1 with step-by-step prompting of the $C^2P$ framework and few-shot learned $C^2P$, significantly improve accuracy in scenarios involving natural stories.

Table 3: Comparison of applying C$^2$P frameworks with existing LLMs with CoT in responding to queries within natural stories, TP: True Positives, FP: False Positives, TN: true negatives, FN: False Negatives

| Models | TP | FP | TN | FN |
|---|---|---|---|---|
| **LLMs without C$^2$P** | | | | |
| GPT-3.5 | 5 | 8 | 7 | 10 |
| GPT-4 Turbo | 6 | 7 | 8 | 9 |
| GPT-4 | 7 | 8 | 7 | 8 |
| GPT-4O | 8 | 6 | 9 | 7 |
| LLaMA 3-70B | 9 | 9 | 6 | 6 |
| LLaMA 3.1-70B | 9 | 10 | 5 | 6 |
| **Featured LLMs with C$^2$P** | | | | |
| **C$^2$P on LLaMA 3.1-70B** | **11** | **3** | **12** | **4** |
| **C$^2$P Few-shot learned LLaMA 3.1-70B** | **10** | **3** | **12** | **5** |
| **C$^2$P on GPT-4 Turbo** | **13** | **1** | **14** | **2** |
| **C$^2$P Few-shot learned GPT-4 Turbo** | **11** | **3** | **12** | **4** |

Additionally, Table 4 provides an example of such natural stories and demonstrates the result of applying few-shot learned GP4-Turbo and LLaMA 3.1 with C$^2$P to them. This showcases how GPT models (including versions 3.5, 4, 4 Turbo, and 4O) fail to respond correctly. It is important to note that the few-shot learning process of GPT-4 Turbo and LLaMA 3.1 was performed using only ten examples from the CORR2CAUSE dataset, without including any natural story examples.

Table 4: Example of a natural story and the result of few-shot learned GPT-4-Turbo with C$^2$P

| **Example of true story** | **Results of few-shot learned GPT-4 Turbo with C$^2$P** |
|---|---|
| **Premise**: eating junk food (A), obesity (C), and watching television (B) have relations with each other. There is a correlation between eating junk food and obesity, and between watching television and obesity. However, eating junk food and watching television are independent from each other. **Hypothesis**: Eating junk food directly affects obesity | Step 1: `Number of random variable:3` `Names of random variable:  A, B, C` |
| | Step 2: `All of Statistical Relations:{` `Dependencies:  [[A, C], [B, C]],` `Unconditional Independencies:  [[A, B]],` `Conditional Independencies:  []}` |
| | Step 3: `A: {A: 0, B: 1, C: 1},` `B: {A: 1, B: 0, C: 1},` `C: {A: 1, B: 1, C: 0}` |
| | Step 4: `A: {A: 0, B: 0, C: 1},` `B: {A: 0, B: 0, C: 1},` `C: {A: 1, B: 1, C: 0}` |
| | Step 5: `A: {A: 0, B: 0, C: 1},` `B: {A: 0, B: 0, C: 1},` `C: {A: 1, B: 1, C: 0}` |
| | Step 6: `C: [A, B]` |
| | Step 7: `C: [A, B]` |
| | Step 8: `A: {A: 0, B: 0, C: 1},` `B: {A: 0, B: 0, C: 1},` `C: {A: 0, B: 0, C: 0}` |
| | Step 9: `Checking matrix[A][C] = 1 and matrix[C][A] = 0.` `According to rule 2, this suggests` A is a direct cause of C, or C is a direct effect of A. |

## 4.5 Evaluation of the C$^2$P framework on the (co)evolution of supermassive black holes and their host galaxies

In astrophysics, experiments are impossible. Thus, causal reasoning must rely exclusively on observational data. Pasquato et al. (2023) computationally studied the coevolution of supermassive black holes based on numerical data, and a causal graph for the underlying mechanism is extracted. Then, the causal hypothesis, "central density affects black hole mass", is answered. Similar to their study, we (co)evolve supermassive black holes (SMBHs) and their host galaxies based on the verbalized information in (Pasquato et al., 2023). We GPT-4 Turbo with C$^2$P to process the verbalized information provided in subsection 4.1, obtain a PDAG, and then reason about the causal questions.

Fig. 4 illustrates the results of the subtasks of C$^2$P on SMBHs data presented. The first four subtasks aim

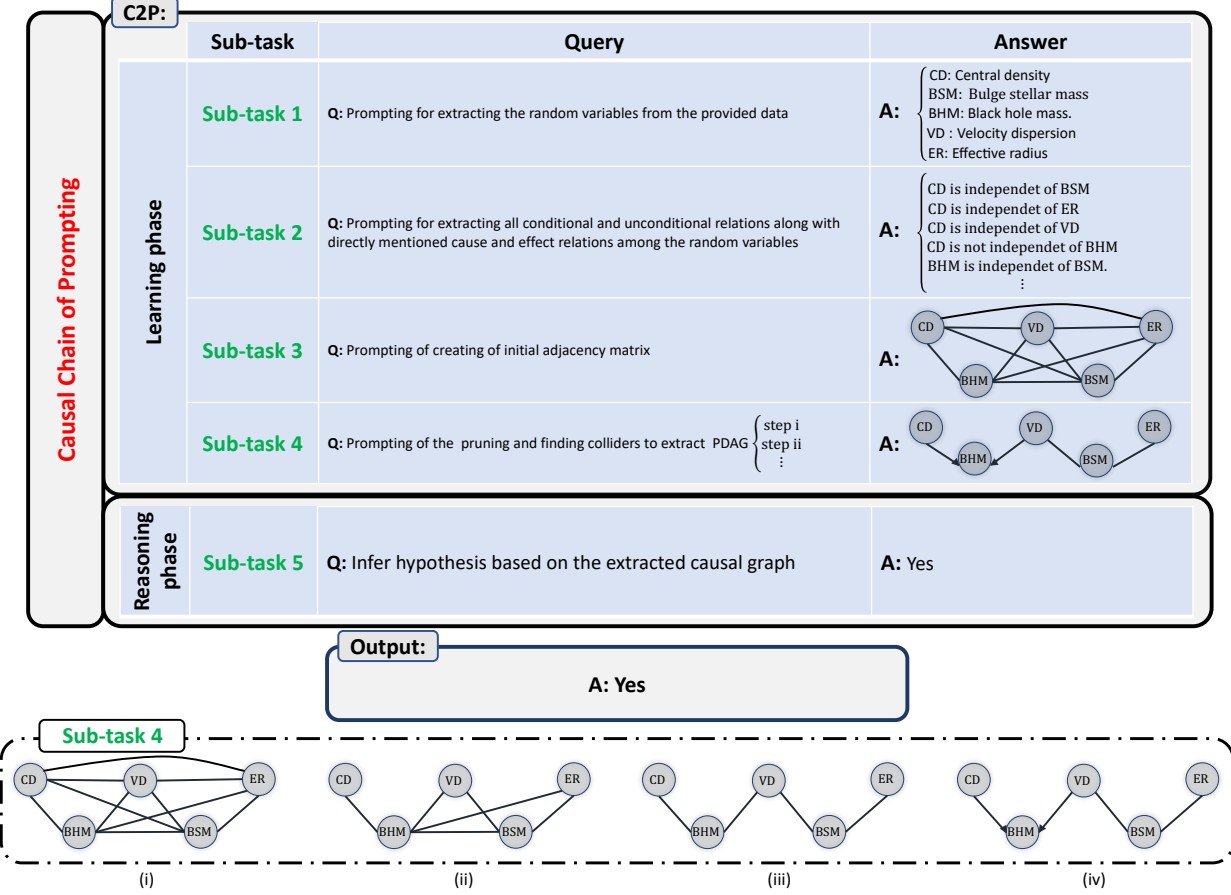

Figure 4: Prompts (**Q**) and results (**A**) of subtasks application of the C$^2$P framework to real-world complex scenarios and steps of subtask 3 for the given premise.

to extract the PDAG from the given data. Then, the hypothesis that central density affects black hole mass is evaluated in subtask 5 and answered. According to the extracted PDAG, many other reasoning questions questions can be answered that are too complicated for the existing LLMs to answer. It is important to note that by implementing C$^2$P, not only many causal questions can be answered with this approach, but it is easy to show which causal questions can not be answered based on the given premise and which more information has to be given to be able to causally reason. The discussion on which questions can and cannot be answered based on PDAG is presented in detail in studies such as (Hernán & Robins, 2006; Hauser & Bühlmann, 2012; Perkovic, 2020). Importantly, the inability to answer certain causal reasoning questions based on the PDAG, or its corresponding adjacency matrix, does not reflect a limitation of the C$^2$P framework. Rather, any rational agent needs additional information to effectively address all causal inquiries.

## 5    Discussion, Challenges and Future Works

**Practical insights on simulations and results of $C^2P$:** Each part of our study has a specific aim. The goal of the simulations, where we applied the 9 steps of $C^2P$ step by step, is to show what the expected output would be and how it leads to causal reasoning. However, our investigations demonstrate sensitivity to the prompts. In few-shot learning, where examples of expected results are provided for each prompt, the sensitivity to prompts was not significant; instead, the results were primarily dependent on the number of examples. Interestingly, while the prompt is lengthy —due to the fact that all steps are given to LLMs as a single prompt —- the results remained highly accurate with as few as ten examples, as reported in Tables 1 and 4. By comparing the prompts in Appendices A.3 and A.4, it is clear that the prompts in the few-shot learning section do not need to be as detailed and are more abstract than those used in the step-by-step application of $C^2P$. Additionally, it is important to note that the results for natural stories using few-shot learned $C^2P$ were highly accurate (see Table 4), even though the examples in the few-shot learning process were synthetic "cause-effect" examples from the CORR2CAUSE dataset rather than real story examples.

The comparison of results between $C^2P$ and CoT provides new insights into how LLMs aim to reason and why $C^2P$ and similar frameworks can guide a model to "think" step by step. As demonstrated in the examples provided in the code repository—under "Sample Responses of LlaMa 3.1 with CoT" for LlaMa 3.1 and "Sample Responses of GPT-4 Turbo with CoT" for GPT-4 Turbo— both models attempt to break tasks into multiple subtasks when they are asked to think step by step (sometimes with more than nine consecutive subtasks). However, they still lack rationality in how they structure these subtasks, as it is discussed in Wei et al. (2022a). In other words, they struggle to design the necessary subtasks for reasoning, a task that $C^2P$ successfully performs to enhance reasoning. This challenge is also highlighted in Wei et al. (2022b), where CoT is introduced, and its logical inconsistencies and poor step alignment are discussed as key limitations of the approach.

**Causal reasoning and identifiability:** Extracting a PDAG using only observational data is a key step in learning the true underlying causal mechanism. Based on the extracted PDAG, two main questions arise: Given a PDAG, under what conditions can we make causal reasoning? This involves determining the necessary assumptions and data required to address a causal question. This issue is known as identifiability. It is generally possible to answer some cause-effect questions based on the PDAG, represented by a causal graph, and lower-level data. These questions are mainly the ones that are related to the part where we have extracted a collider, or if the question is just about the existence of a cause-effect relation; in some cases, these questions can be answered as well. However, it is almost impossible to answer all the causal questions based on the given information, as discussed in (Bareinboim et al., 2022; Pearl & Bareinboim, 2022). As a result, the second question is, what else is needed to answer all the causal questions? In such cases, interventional data is necessary to explicitly extract all the directions of causal relations in a mechanism. To do so, while a naive baseline approach would require $O(n^2)$ interventions, various methods have been proposed, such as those in (Kocaoglu et al., 2017; Choo & Shiragur, 2023; von Kügelgen et al., 2024; Squires & Uhler, 2023), for cases in different situations. However, one clear thing is that it is not possible to reason all the causal questions. For instance,

- Directions cannot be discovered: PDAGs typically include both directed and undirected edges. The undirected edges indicate uncertainty about the direction of causality. It is important to note that without further data or assumptions, it is not possible to definitively determine the causal direction for these relationships with any method.

- Full causal path analysis: While PDAGs can indicate possible paths between variables, they may not fully reveal which paths are indeed causal and which are due to confounding or indirect effects. Questions about specific causal pathways can thus be hard to answer definitively.

- Predictions under interventions: Questions about the outcomes of hypothetical or actual interventions on one or more variables (do-calculus questions) often require a fully specified causal model.

PDAGs, with their partial specification, might not support detailed predictions under interventions without resolving the ambiguities in causal direction.

**Next Steps for Achieving Reasoning in LLMs:** The human brain possesses an intrinsic drive to understand causality. Whether driven by curiosity or the pursuit of specific goals, we continuously seek to understand why events occur and how they are interconnected. Causal reasoning is a broad and complex task in AI and LLMs. While current machine learning methods find it difficult to extract causal structures and subsequently reason causally, this problem can be even more intricate within the context of LLMs. The primary reason for this complexity is the distinction between association and cause-effect relationships. The inherent structure of LLMs relies on the attention mechanism presented in (Vaswani et al., 2017), which is akin to what Pearl refers to as "association" in the ladder of causality (Pearl & Mackenzie, 2018). As a result, advancing up the ladder of causality is essential for enabling true reasoning. To enable LLMs to address this fundamental quest for causality, they must be capable of sequentially performing two main tasks. As demonstrated in our study, $C^2P$ has significantly enhanced the reasoning capabilities of LLMs on the provided datasets, which can be the main task in the reasoning process. However, the first task is to be able to understand the causal questions, which involves formalizing the definition of causal questions and establishing a taxonomy for finer-grained classification. The initial effort on this is presented in Ceraolo et al. (2024). Then, $C^2P$ can be employed. However, the first subtask of this framework is to extract random variables, which requires the models to understand the definition of a random variable to ensure that the variables in the reasoning process are correctly identified. Failure in this process could undermine our proposed framework. Another important concept in this process is extracting the dependencies provided in the premise. In the datasets used for our experiments, these dependencies are explicitly stated in the given premise; however, in reality, they are often implicit. As a result, LLMs should be able to accurately extract these dependencies. Additionally, more comprehensive examples and scenarios need to be generated to aid in the learning process of an LLM. By overcoming these challenges, the integration of $C^2P$ with LLMs can provide these models with causal reasoning capabilities, similar to the transformative impact of "Chain-of-Thought" (Wei et al., 2022b), as highlighted by Chung et al. (2024). It is also important to note that while our experimental results with few-shot learned LLMs show a significant increase in accuracy for tasks involving direct cause-and-effect relationships or common cause questions, the models still struggle with indirect cause-and-effect questions. Moreover, the accuracy of our framework decreases as the number of random variables increases. As Pearl demonstrated that DAGs and d-separation are complementary in causal reasoning tasks, more structured subtasks can be introduced to improve performance in answering indirect causal questions and handling more complex scenarios with numerous variables. These subtasks would help guide LLMs in performing reasoning tasks more effectively.

The key benefit of Few-shot learning is that it demonstrates that pre-trained models can perform causal reasoning without needing to be retrained from scratch or necessarily fine-tuned, making it more cost-effective and compute-efficient. This leads to reduced costs and less demanding infrastructure requirements. Therefore, a model fully trained or fine-tuned with $C^2P$ would probably perform even better than few-shot learned ones, as it can be trained or fine-tuned on thousands of examples instead of the ten examples used during few-shot learning. This highlights how integrating $C^2P$ during the training and fine-tuning of LLMs can revolutionize existing models. This is also discussed in Chung et al. (2024) as the "scaling law". Additionally, Studies like Kaplan et al., 2020 showed that language models benefit from increased data and computational resources. More data during fine-tuning allows the model to capture better representations, leading to improved performance in various tasks. This is one of the core principles behind why larger datasets, would likely boost accuracy compared to just 10 examples. In Brown (2020) with GPT-3, the authors highlight how performance increases as the model is trained on more examples. In few-shot learning, the model can generalize from a small set of examples, but fine-tuning with larger datasets usually leads to more substantial improvements. Devlin (2018) also demonstrates that fine-tuning with more task-specific examples improves the model's performance. As a result, by addressing the mentioned challenges, employing $C^2P$ in the fine-tuning process is expected to improve the reasoning accuracy of the models.

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

# A    Appendix

## A.1    do-calculus rules

The three main rules of the do-calculus are as follows:

1. **Insertion/deletion of observations**: If a variable $X_2$ is independent of $X_1$ given a set of other variables $X_3$, then the conditional distribution of $X_2$ given $do(X_1)$ and $X_3$ is the same as the conditional distribution of $X_2$ given $X_3$ alone. This rule can be applied when $X_1$ and $X_2$ are d-separated by $X_3$ in the graph modified by removing all arrows pointing into $X_1$.

2. **Action/observation exchange**: If $X_2$ is independent of $X_1$ given $X_3$ and another set $W$ where $X_1$ is intervened, then the graph can be modified to reflect this exchange.

3. **Insertion/deletion of actions**: This rule states that if intervening on $X_1$ does not affect $X_2$ after accounting for $X_3$, the intervention can be ignored in the expression.

## A.2    Example of comparison in responding to causal queries

For the given example in Subsection 4.1, Table 5 demonstrates that while GPT-4 arrives at the correct final answer, the reasoning process is flawed, providing further evidence of the random behavior of existing models in causal reasoning.

Table 5: An example of the correct final answer with GPT-4 Turbo while the reasoning process leading is incorrect and results of C$^2$P Few-shot learned GPT-4 Turbo

| GPT-4 Turbo with CoT | C$^2$P Fewshot learned GPT-4 Turbo |
|---|---|
| The hypothesis that A directly affects C could be possible. The premise states that A correlates with C, which means there is a relationship between them. However, correlation does not necessarily imply causation. It could be that A affects C, C affects A, or that both A and C are affected by a separate, unmentioned variable. However, since the premise does not provide any information contradicting the hypothesis, it is possible that A directly affects C. Final Answer: "Yes" | According to the adjacency matrix, matrix["A"]["C"] = 1 and matrix["C"]["A"] = 0 indicates that A directly cause C. Therefore, the hypothesis that A causes C or C is effect of B is correct. Final Answer: "Yes" |

## A.3    Prompts for C$^2$P frameworks

The proposed steps (prompts) for step-by-step proceeding with 5 main subtasks of C$^2$P are as follows:

1. **Prompting for extracting the random variables from the provided data.**

   - **Step 1:** Please give the number of random variables in the given premise and write the names of all random variables.

2. **Prompting to extract all the cause and effect relations along with all conditional and unconditional relations among the random variables specifically mentioned in the given premise**

   - **Step 2:** If 2 random variables, for instance, $R_1$ and $R_2$, are independent, write it in this form: "$R_1$ is independent of $R_2$". If there exist 2 random variables, for instance, $R_1$ and $R_2$, are

conditionally independent given a third random variable, for instance, $R_3$, write it in this form: "$R_1$ and $R_2$ are independent given $R_3$". If two random variables, for instance, $R_1$ and $R_2$, are specially mentioned to have cause and effect relation, write it in this form: "$R_1$ is the cause of $R_2$".

3. **Prompting to create an adjacency matrix where all elements are 1, except for the diagonal elements and the elements corresponding to the cause-and-effect relationships specifically mentioned in the given premise.**

   - **Step 3:** In this phase, each random variable is treated as a node within a fully connected undirected graph. Then, for each pair, for instance, $R_1$ and $R_2$, presented in the form: "$R_1$ is the cause of $R_2$", set the element in ["$R_2$", "$R_1$"] in the adjacency matrix to 0.

4. **Prompting of the conditional and unconditional independency valuation and identifying the colliders, step by step, to extract the causal PDAG.**

   - **Step 4:** Update the adjacency matrix based on the specified unconditional independencies between random variables. Each pair of variables that is declared independent should have their corresponding value set to zero in the adjacency matrix. The initial adjacency matrix and the list of independencies are provided below. Please ensure all independencies are correctly reflected in the updated matrix. Instructions: - For each pair of variables listed as independent, set their corresponding entries in the adjacency matrix to 0.

   - **Step 5:** Update the adjacency matrix based on the specified conditional independencies between random variables. Each pair of variables that is declared independent should have their corresponding value set to zero in the adjacency matrix. The initial adjacency matrix and the list of independencies are provided below. Please ensure all independencies are correctly reflected in the updated matrix. Instructions: - For each pair of variables listed as independent given other variable(s), set their corresponding entries in the adjacency matrix to 0.

   - **Step 6:** Task: Given an initial adjacency matrix, follow these steps: Step 1: Identify all rows (key values) in the matrix where there are two or more than two columns with the value "1" in them. For each identified row, find all pairs of different columns where the values are "1".Ensure to exclude rows that do not contain any pairs from the results. Step 2: Display these pairs, "All Pairs", where each row name is key, and the value is a list of column names that are identified in Step 1.

   - **Step 7:** Given the "All Pairs" and the list of independencies, follow these instructions step by step: Instruction: For each key in "All Pairs", delete all the pairs that are not mentioned as independent in the "independencies" list and return other with all their values. The "All Pairs" contains pairs of elements associated with each key. The goal is to update this by removing pairs that are not mentioned as independent. The list of independencies provides information about which pairs are independent of each other.

   - **Step 8:** Given the initial adjacency matrix represented and the "All Pairs" list, for each key-value pair ("$R$") in "All Pairs", modify the initial adjacency matrix as follows: 1- Set the value in the "$C_1$" row and "$R$" column to 0: ("$C_1$", "$R$") = 0. 2- Set the value in the "$C_2$" row and "$R$" column to 0: ("$C_2$", "$R$") = 0.

5. **Prompting for cause-and-effect questions or hypotheses**

   - **Step 9:** To extract and understand causal relations in the adjacency matrix: For each specified variable "$R$" and "$C$", for instance, that are listed in the adjacency matrix: - If matrix entry at [ "$R$", "$C$"] = 1 and ["$C$", "$R$"] = 1, then the causal direction between "$R$" and "$C$" is undetermined. - If matrix entry at ["$R$", "$C$"] = 1 and ["$C$", "$R$"] = 0, then "$R$" is a direct cause of "$C$" or "$C$" is a direct effect of "$R$". - If matrix entry at ["$R$", "$C$"] = 0 and ["$C$", "$R$"] = 1, then "$C$" is a direct cause of "$R$" or "$R$" is a direct effect of "$C$". If two variables directly affect a third variable, the first two variables are common causes, and the third variable is a collider. Evaluate the hypothesis based on the given partially presented as an adjacency matrix with the given Instruction.

## A.4  Prompts for few-shot learning of C²P

An example of given prompts for the few-shot learning process of C²P is as follows:

**Premise**: Suppose there is a closed system of 5 variables, A, B, C, D, and E. All the statistical relations among these 5 variables are as follows: A correlates with C. A correlates with D. A correlates with E. B correlates with D. B correlates with E. C correlates with D. C correlates with E. D correlates with E. However, A is independent of B. A and B are independent given C. B is independent of C. B and C are independent given A. C and E are independent given A, B, and D.

**Hypothesis**: There exists at least one collider (i.e., common effect) of A and B.

- **"Subtask 1"**- The number of random variables and their names in the given premise in JSON format:

  **Output**: "number of random variables: 5, "names of random variables": ["A", "B", "C", "D", "E"]

- **"Subtask 2"**- All the dependencies, conditional and unconditional independencies between all random variables extracted in "subtask 1":

  **Output**: {"All of Statistical Relations": {"Dependencies": [["A", "C"], ["A", "D"], ["A", "E"], ["B", "D"], ["B", "E"], ["C", "D"], ["C", "E"], ["D", "E"]], "Unconditional Independencies": [["A", "B"], ["B", "C"]], "Conditional Independencies": [["A", "B"], ["B", "C"], ["C", "E"]]}}

- **"Subtask 3"**- The adjacency matrix of all random variables extracted in "subtask 1" where each random variable is treated as a node within a fully connected undirected graph:

  **Output**: {"A": {"A": 0, "B": 1, "C": 1, "D": 1, "E": 1}, "B": {"A": 1, "B": 0, "C": 1, "D": 1, "E": 1}, "C": {"A": 1, "B": 1, "C": 0, "D": 1, "E": 1}, "D": {"A": 1, "B": 1, "C": 1, "D": 0, "E": 1}, "E": {"A": 1, "B": 1, "C": 1, "D": 1, "E": 0}}

- **"Subtask 4"**- Update the adjacency matrix extracted in the output of "subtask 3" based on the specified unconditional independencies between random variables. Each pair of variables that are declared independent should have their corresponding value set to zero in the adjacency matrix. - For each pair of variables listed as unconditional independent in "subtask 2", we set their corresponding entries in the adjacency matrix to 0. - We do not change any other entries except those specified by the independence.

  **Output**: {"A": {"A": 0, "B": 0, "C": 1, "D": 1, "E": 1}, "B": {"A": 0, "B": 0, "C": 0, "D": 1, "E": 1}, "C": {"A": 1, "B": 0, "C": 0, "D": 1, "E": 1}, "D": {"A": 1, "B": 1, "C": 1, "D": 0, "E": 1}, "E": {"A": 1, "B": 1, "C": 1, "D": 1, "E": 0}}

- **"Subtask 5"**- Update the adjacency matrix in the output of "Subtask 4" based on the specified conditional independencies between random variables extracted in "subtask 2". Each pair of variables that are declared conditional independent should have their corresponding value set to zero in the adjacency matrix. - For each pair of variables listed as conditionally independent given other variable(s), we set their corresponding entries in the adjacency matrix to 0. - We do not change any other entries except those specified by the conditional independence.

  **Output**: {"A": {"A": 0, "B": 0, "C": 1, "D": 1, "E": 1}, "B": {"A": 0, "B": 0, "C": 0, "D": 1, "E": 1}, "C": {"A": 1, "B": 0, "C": 0, "D": 1, "E": 0}, "D": {"A": 1, "B": 1, "C": 1, "D": 0, "E": 1}, "E": {"A": 1, "B": 1, "C": 0, "D": 1, "E": 0}}

- **"Subtask 6"**: In this subtask, for each key value in the "Adjacency Matrix" mentioned in the output of "Subtask 5", the task is to extract only pairs of columns with values "1" step by step according to the provided steps and put it in "Candidates" dictionary: follow these steps: Step 1: Identify all rows (key values) in the matrix where there are at least two or more columns with the value "1". For each identified row, find all pairs of different columns where the values are "1". Ensure to exclude rows that do not contain any pairs from the results. Make sure to check each row individually and include all valid pairs for every row. Step 2: Display these pairs in a simplified JSON format, where

each row name is a key, and the pair values are a list of column names that are identified in step 1. If there are no such rows, show an empty JSON object. Step 3: Do not include any rows with one or zero columns with "1" values in the output. Example output format: Candidates: { "row1 name": [["1value column1 name", "1value column2 name"], ["1value column1 name", "1value column3 name"], ["1value column2 name", "1value column3 name"]], "row2 name": [["1value column5 name", "1value column6 name"]], ... } The values in each row have to be pairs and the output cannot be as follows: { "row1 name": [["1value column1 name"], ["1value column1 name"], ["1value column2 name", "1value column3 name"]], "row2 name": [["1value column5 name"]], ... } Please provide only the desired output formatted exactly as shown in the example without any further explanation.

**Output**: {"A": [["C", "D"], ["C", "E"], ["D", "E"]], "B": [["D", "E"]], "C": [["A", "D"]], "D": [["A", "B"], ["A", "C"], ["A", "E"], ["B", "C"], ["B", "E"], ["C", "E"]], "E": [["A", "B"], ["A", "D"], ["B", "D"]]}

- **"Subtask 7"**- In this subtask, given the output of "Subtask 6", "Candidates", and "Unconditional Independencies" in "Subtask 2", the task is to identify and extract all the pairs in the "Candidates" that are also present in the "Unconditional Independencies" step by step according to the provided steps. follow these steps: 1- For each pair in the "Candidates" list, check if it is present in the "Unconditional Independencies" list. 3-Only keep all the pairs from "Candidates" that are also present in "Unconditional Independencies". If a pair in "Candidates" is found in "Conditional Independencies", keep it. 4-Remove any pairs in "Candidates" that are not found in "Conditional Independencies". If a pair in "Candidates" is not found in "Unconditional Independencies", remove it. 5-Output the result as the modified "Candidates" dictionary without any additional text or explanation. Only the updated "Candidates" dictionary and nothing else.

  **Output**: {"D": [["A", "B"], ["B", "C"]], "E": [["A", "B"]]}

- **"Subtask 8"**- Given the adjacency matrix in the output of "Subtask 5" and the "All Pairs" list in the output of "Subtask 7", for each key-value pair ("R") in "All Pairs", we modify the adjacency matrix as follows: -For each key "R" and pair ("C1", "C2") in the candidates, change the values in key "R" and pairs ("C1", "C2") to zero. -Ensure that only the specified modifications are made, and all other entries in the adjacency matrix remain unchanged.

  **Output**: {"A": {"A": 0, "B": 0, "C": 1, "D": 1, "E": 1}, "B": {"A": 0, "B": 0, "C": 0, "D": 1, "E": 1}, "C": {"A": 1, "B": 0, "C": 0, "D": 1, "E": 0}, "D": {"A": 0, "B": 0, "C": 0, "D": 0, "E": 1}, "E": {"A": 0, "B": 0, "C": 0, "D": 1, "E": 0}}

- **"Subtask 9"**- In this subtask, given the "Adjacency Matrix" mentioned in the output of "Subtask 8", evaluate the correctness of the hypothesis step by step according to the provided rules. Rules: 1- If matrix["R"]["C"] = 1 and matrix["C"]["R"] = 1, then the causal direction between "R" and "C" is undetermined. 2- If matrix["R"]["C"] = 1 and matrix["C"]["R"] = 0, then "R" is a direct cause of "C", or "C" is a direct effect of "R". 3- If matrix["R"]["C"] = 0 and matrix["C"]["R"] = 1, then "C" is a direct cause of "R", or "R" is a direct effect of "C". 4- Test each variable in the matrix, for instance, "R", "R" is a collider if the matrix entries in ["R", "C1"] = 0, ["C1", "R"] = 1, ["R", "C2"] = 0, and ["C2", "R"] = 1 Perform it step by step and provide the final "Yes" or "No" answer.

  **Output**: Step 1: According to the rules, a node is a collider (common effect) if the matrix entries in ["R", "C1"] = 0, ["R","C2"] = 0, ["C1", "R"] = 1 and ["C2", "R"] = 1.

  Step 2: According to rule 4, "R" is a collider if the matrix entries in ["R", "D"] = 0, ["R", "E"] = 0, ["D", "R"] = 1 and ["E", "R"] = 1.

  Step 3: Checking each variable in the matrix

  Node C: ["C", "A"] = 1 and ["A", "C"] = 1, so C is not a common effect of A and B.

  Node D: ["D", "A"] = 0 and ["A", "D"] = 1, and ["D", "B"] = 0 and ["B", "D"] = 1, so D is a common effect of A and B.

  Node E: ["E","A"] = 0 and ["A","E"] = 1, and ["E","B"] = 0 and ["B", "E"] = 1, so E is a common effect of A and B.

  Step 4: So, the hypothesis that there exists at least one collider (i.e., common effect) of A and B is correct. The answer is "Yes".

