# OpenReview forum: "$\text{C}^\text{2}\text{P}$: Featuring Large Language Models with Causal Reasoning"
_TMLR — Rejected by TMLR_

### Review · Reviewer_3jUT · 2024-08-30

**Summary Of Contributions:**

This paper introduces C2P (Causal Chain of Prompting), a prompting framework to help language models to solve causal reasoning tasks.
C2P consists of several steps, such that LLMs extract the relevant variables from a task and generate an adjacency metrics based on the relation among variables.
Performing these steps, consisting of 9 prompts, GPT 4-Turbo was able to achieve accuracy improvements of 33% in contrast to models prompted without C2P.
C2P is evaluated on a subset of the Corr2Cause dataset, as well as a newly created datasets with more natural sounding problems.
Both the standard and few-shot variants of C2P have achieved performance improvements.

**Audience:**

Yes

**Broader Impact Concerns:**

No concerns

**Claims And Evidence:**

No

**Requested Changes:**

- Rewrite Section 2: At the moment it appears that “Markov property” and “Markov Equivalence of Graphs” have been copied from (Jin et al., 2023b)

- Adjust claims or provide more details/insights
  - Usefulness for fine-tuning
  - Naturalness of the newly created dataset
  - Ability to enhance multiple LLMs (bottom of Page 10)


- Extend the baselines used for experiments
  - Provide more details on the random baselines and why their performance is better than 50%
  - Add chain of thought baseline
  - Add GPT-4 Turbo (without C2P) to Table 1 and 2
  - Apply C2P to other models listed in Table 1
  - Potentially perform an ablation study to investigate whether every step of the framework is needed (e.g., get solutions after step 2)

- Share more details in the online repository
  - Scripts to use LLMs
  - Corr2Cause subset of 100 instances
  - Reponses from LLMs
  - Outcomes/labels for the natural stories


Small adjustments:
- Section 4.1. Make sure that the references to Llama3 and the year of accessing GPT4 are correct. If GPT4 was used in April 2023, it would be good to redo the experiments with the current version
- PC algorithm page 4 uses A,B,C instead of x
- “Various benchmark datasets” provide the exact number
- Page 3 bottom “The be able”
- Section 5 misses references (“as indicated by prior research ().”)
- Table 3 refers to GPT4.5 rather than 3.5
- Unclear what “featuring” means in Section 4 title
- The first couple of sentences in Section 5 are difficult to follow for me
- State the exact number of samples in the newly created dataset

**Strengths And Weaknesses:**

**Strengths**
- Elaborate prompting procedure
- Good overview and introduction to causal reasoning
- Well-written and helpful visualisations
- Performance improvements with standard and few-shot setting
- Not relying on external tools
- Consider data contamination



**Weaknesses**

**Overclaiming:**
The newly created dataset is called "natural" while Corr2Cause is named "synthetic". I am not certain that such a distinction can be made, as both seem to be created following similar .
Moreover, this dataset of 10 instances is used to state that "C2P enhances LLMs’ ability to
causally reason in real-world scenarios, addressing complex problems in fields such as healthcare, medicine, economics, education, social sciences, environmental science, and marketing".
This seems like very strong claims given 10 samples.
The subsampling of the Corr2Cause dataset is stated to "enhances both the realism and comprehensibility of the results.”. While it balances the dataset, the impact on realism and comprehensibility is not as clear.

The third contribution aims to " show how integrating C2P during the training or fine-tuning of LLMs can revolutionize existing
models". However, there is no fine-tuning peformed or empirical evidence. It is only hypothesized that this can be true in Section 5.
I would suggest to remove this claim or add experiments to investigate this.


**Empirical validation:**
While the empirical validation considers several models and datasets, there are some shortcoming. For once, C2P is only applied to GPT-4 Turbo and none of the other LLMs. Additionally, the default performance of GPT-4 Turbo is not shown, which makes a direct comparison to determine the performance improvement more difficult.
In terms of baselines, it would be helpful to see the effect of Chain of thought prompting in addition to random baselines.
The random baselines perform quite well, and some details on how they work would be interesting. In particular, as they exceed 50% accuracy.
Lastly, the two datasets investigated are relatively small. It is not clear why Corr2Cause was reduced to 100 instances from 200.000.
This could have impacts on the validity, given that even at a temperature of 0, LLMs can be non-deterministic.


**Shared resources**:
The repository contains a prompt and the created dataset. However, more information is needed to allow for reproducibility.
It would be helpful to have access to the code used to run the LLMs.
Also, the 100 sample subset from Corr2Cause should be shared.
If possible, the responses from the LLMs should be shared as well, such that the results can be verified.

---

> ### Author Response · Authors · 2024-10-08
> **We thank the reviewer for their comments, which we address in order. Due to the character limitation, we briefly explain our idea for your feedback and detailed the revisions made to the manuscript, in response to your comments. We hope that we have adequately addressed all of them.**
>
> Overclaiming: In Jin et al. (2023b), two datasets, the synthetic data and "Natural Story" that we used both of them. In the revised version, we used 30 "Natural Story" in our experiments to better evaluate our method. We have added subsection 4.1 to better explain the data we used in our experiments.  We have rewritten the third contribution and also added a paragraph in the discussion section, "Why does integrating ...",  to give a better understanding of the presented contribution.
>
> Empirical validation:   The goal in corr2cause was to provide a benchmark dataset, and the responses in their data are mostly  "No". A limitation of their study is that metrics (F1, recall, ...) do not transparently highlight the differences between models, as they noted. To address this, we randomly selected samples from their dataset while ensuring that the "Yes" and "No" responses were balanced. This offers a clearer understanding of the current models' performance based on 4 metrics. We have employed 120 samples in our examples for synthetic data and 30 samples for "natural study." Responses to some of our simulations are added to the code repository to give a better understanding of LLMs and the framework. The aim of our study is to prove the concept, so, for data number and to evaluate the significance of improvement, we have used the sample size formula for comparing two proportions and showed that with 117 examples, the improvement is significant with 99% confidence. In Table 1, in the last version of the paper, the results for LlaMa were not with zero temperature which is corrected in the new version.
>
> Shared resources: We have updated the code repository.
>
> Rewrite Section 2: We have revised them in the revised version.
>
> Adjust claims or provide more details/insights:
> - The key benefit of few-shot learning is that it shows pre-trained models can perform causal reasoning without the need for retraining from scratch, making the process more cost-effective and compute-efficient. This results in reduced costs and less demanding infrastructure requirements. Additionally, we aim to demonstrate that by performing few-shot learning with only 10 examples, fine-tuned models with a higher number of examples with $C^2P$ reason more accurately than existing models. Consequently, a model fully trained with $C^2P$ would likely perform even better, as it could be trained on thousands of examples instead of the 10 used during fine-tuning.
>
> -In Jin et al., 2023b, two main datasets are presented: the corr2cause data and "Natural Story". We used the same process to generate the data and verify their correctness. In the revised version, we added subsection 4.1 to clearly clarify the data used in our simulations and experiments. Additionally, details about these datasets are provided in the code repository, which includes both the data creation process and example outputs.
>
> -We have added the results of $C^2P$ on LLaMA 3.1 to demonstrate the generalizability of our method and provide further evidence to support our claims. Our goal was to show that this framework can be applied to other LLMs to enhance reasoning. These results have been included in the new version, and the corresponding sentence has been revised accordingly.
>
> Extend the baselines used for the experiment
> - First, we have to note that while a model can be worse than others in one metric, it can be better than them in other metrics. This is one of the main reasons that we sampled the data from corr2cause dataset to be balanced in "YES" and "NO". Based on the results reported in our code repository, we believe the main reason for this is the hallucinations and randomness in LLM. These details have been incorporated into the revised version of the manuscript, presented after Table 1.
> - The chain of thought approach has already been applied to all the LLMs, and the prompts along with their results, are available in the code directory.
> - Thank you for pointing that out. It is included in the new version.
> -We applied $C^2P$ to LLaMA 3.1 to demonstrate the generalizability of our method. Due to token limitations, both CoT and C^2P could only be applied to LLaMA 3.1. While we can apply the nine main prompts to LLaMA 3, few-shot learning is not feasible because of these token constraints. We have noted that our framework, like most frameworks for LMs in various tasks, requires a certain amount of tokens.
> - Causal reasoning is a complex task, with each step playing a crucial role in the reasoning process. As presented in five subtasks, or equivalently, nine prompts, each step builds on the results extracted from the previous subtasks or prompts. Consequently, omitting any step leads to erroneous outcomes.
>
> Share more details in the online repository.
> We have updated it.
>
> Small adjustments:
> We addressed all the mentioned comments in the new version. "featuring" means to include something as an important part
>  However, if the title seems inappropriate, further revisions can be made.

---

> > ### Comment · Reviewer_3jUT · 2024-10-11
> > **Response to comments**
> >
> > Thanks a lot for the response, which clarifies some of my concerns.
> > There are still some aspects which I would like to clarify.
> >
> > First, with regards to fine-tuning. You state that the third contribution is rewritten and also that fine-tuning “would likely perform better”.
> > This still lets me to believe that such a claim should not be made without experimental proof. To what degree has the text been rewritten and does is this statement still made?
> >
> > The code repository was updated with some details, but I still miss the responses generated by the model. At the moment there only are sample responses from Llama but none of the GPT models.
> > Moreover, the COT results are shown for Llama only with the entire log file. However, there is no summary of the results and other models are missing.
> > Therefore, I would like to ask how COT compares with C2P.
> >
> > To clarify my question regarding the baseline, I am curious about how the two random baselines are implemented.
> >
> > One inconsistency I noticed in the responses is that you stated that the metrics (F1, recall) by Lin et al, are a limitation, but in the response to
> > reviewer-54PX that Jin et al. suggested only F1 score is valid. It is important to clarify this limitation as it is one of the motivating factors to create a balanced subset/dataset.
> >
> > Lastly, in terms of character limitations in the response, it could be an option to respond to reviews with multiple comments

---

> ### Author Response · Authors · 2024-10-12
>
> Thank you so much for your insightful comments.
>
> 1-	In the new version, we omitted this from the contribution section since we are unable to perform an experiment on it.
> We mostly included in the future works subsection.
> We have also updated the code repository by adding a new script, "Generating Few-Shot Prompts.py". This script allows for more examples to be included in the few-shot learning process or replaced with other examples. It currently contains 52 examples, and by increasing the number of examples used (as much as the LLM’s input token limit permits), the accuracy improves. Based on this, we claim that using a higher number of examples—such as 200 in the fine-tuning process—will lead to even greater accuracy. This is also discussed in \citet{chung2024scaling} as the “scaling law”. Additionally, Studies like Kaplan et al. (2020) in "Scaling Laws for Neural Language Models" show that language models benefit from increased data and computational resources. More data during fine-tuning allows the model to capture better representations, leading to improved performance in various tasks. This is one of the core principles behind why larger datasets, would likely boost accuracy compared to just 10 examples. In Brown et al. (2020) with GPT-3, the authors highlight how performance increases as the model is trained on more examples. In few-shot learning, the model can generalize from a small set of examples, but fine-tuning with larger datasets usually leads to more substantial improvements. The "BERT: Pre-training of Deep Bidirectional Transformers for Language Understanding" paper by Devlin et al. (2019) also demonstrates that fine-tuning with more task-specific examples improves the model's performance. The more examples you provide, the better the model adapts to the target task. Though their focus is on supervised fine-tuning, the principle applies similarly to fine-tuning pre-trained LLMs like GPT or LLaMa.
>
> 2-	We have included the sample descriptive responses for both GPT-4 few-shot learning with $C^2P$ and GPT-4 with CoT in the code repository. The reason we hadn’t included all the responses initially is that running all the simulations is necessary to generate the descriptions, as the original code was set up to output only the final answer—“yes” or “no”—to compare LLMs. Please note that only the final answers are used for computing metrics to ensure our study aligns with existing literature. However, we believe the newly added files, along with those included per your last request, offer a better understanding for readers by showing why and how the models behave randomly and how they generate their hallucinated final answers.
>
> 3-	The description of the random baseline is provided in Jin et al. (2023).
>
> “For the random baselines, we use “random (uniform),” which uniformly samples a label (i.e., 50% for each), and “random (proportional),” which samples a label from a Bernoulli distribution proportional to the development set label distribution.”
> Demonstrating that existing LLMs respond randomly was not the primary goal of our study, as this has been extensively covered in various references from multiple perspectives. We simply mentioned it to ensure clarity for readers. We can even omit this part, since we have described our data distribution on subsection 4.1.
>
> 4-	Apologies for any ambiguity in our previous responses. Jin et al. (2023) reported all four metrics but specifically highlighted the use of the F1 score for method comparison due to the distribution of "yes" and "no" answers, which is now discussed in detail in the revised version of the manuscript. Since their focus was primarily on providing a benchmark dataset rather than improving causality methods, this distinction was not a central aspect of their study.
>
> Thank you for highlighting the last comment. We are now able to provide even more information for the other reviewers as well.

---

> > ### Comment · Reviewer_3jUT · 2024-10-15
> > **Response to comments**
> >
> > Thank you for the clarification.
> > There are two aspect I do not feel are fully addressed.
> > First is the comparison with CoT. There are results files in the repository. I am not sure whether this includes results for both datasets. But more importantly, I would be like to see the results in Table 1 and 3, so that one can compare C2P with CoT.
> >
> > The online repository has an additional file with responses for C2P but not for all models. Please clarify if these all responses can be shared.
> >
> > Lastly, on page 7 it states that "generated approximately 30 stories", the "approximately" does not seem to be necessary.

---

> > > ### Author Response · Authors · 2024-10-15
> > >
> > > The results presented in Tables 1 and 3 for all LLMs are based on CoT, except for those where we employed $C^2P$. This is already noted at the end of Section 4.2. In the updated version, we have also clarified this in the table descriptions. Additionally, as requested by another reviewer, we added more discussion comparing our framework with CoT in the second paragraph of Section 5, updated on October 14th.  There is no confidentiality regarding our results, as all codes, datasets, and prompts are provided, making the results easily reproducible. We added the prompt for few-shot learning and the code that allows users to change the examples in the few-shot prompt from the complete set of 50 examples and a new prompt for few-shot learning can easily be generated. However, as mentioned in our previous responses, we only saved the final answers for models. This is the main reason we titled the files "Sample Responses of Few-Shot Learned GPT-4 Turbo with $C^2P$" and "Sample Responses of GPT-4 Turbo with CoT," as we re-ran the simulations per your previous request. The newly added paragraph in the discussion section was inspired by your comment and these two files. However, to include the exact responses for all models using both CoT and $C^2P$, we would need to rerun the tests for all LLMs, which could take some time. Please let us know if including these full responses for each model is necessary.
> > >
> > > We have also revised the paragraph to exclude the term "approximately."

---

> > > > ### Comment · Reviewer_3jUT · 2024-10-16
> > > > **Response to comments**
> > > >
> > > > Thanks again.
> > > > I have missed that the caption for Table 1 and 3 changed.
> > > > I don't think rerunning all models again would be needed, maybe sharing the final answers could already be helpful.

---

> > > > > ### Author Response · Authors · 2024-10-17
> > > > >
> > > > > Thank you for your insightful comments. As mentioned, we only saved and used the final answers. Consequently, we are rerunning the simulations for a sample of examples and saving each model's results in the format "Sample Responses of *** (Only Final Answers).txt," where *** represents the model's name. We have already updated the files for the LLaMa models and will continue adding the GPT models as their simulations are completed. Please note that the computed metrics may differ slightly from those in the manuscript due to changes in the samples and the number of examples used. We hope this addresses all of your concerns. Additionally, these results are easily reproducible by using the API for each model.

---

> > > > > ### Author Response · Authors · 2024-10-20
> > > > > **Further updates based on your comments and those of the other reviewers**
> > > > >
> > > > > The code repository is updated. We have also updated the manuscript and revised it in different sections. We specifically revised the structure of the "Discussion, Challenges, and Future Work" section in the updated version to incorporate your feedback. Your comments were included as challenges and future steps toward achieving reasoning in complex scenarios.
> > > > >
> > > > > We hope that the revisions to the manuscript have clarified the content and addressed the challenges, making it easier for other researchers to build upon our work. Your feedback, along with that of the other reviewers, has significantly enhanced the quality of the study.

---

### Review · Reviewer_54PX · 2024-09-02

**Summary Of Contributions:**

This paper focuses on the problem of standard LLMs being unable to perform causal reasoning. To make LLMs perform better at causal reasoning tasks, the authors propose a solution based on prompt engineering: prompt the model in a step-by-step fashion, inspired by the PC algorithm, to explicitly extract the underlying causal graph of the query (e.g. extract the random variables, list their relations etc.), amounting to a total of 9 prompts having to be completed by the LM. The proposed method is titled "Causal Chain of Prompting" (C2P). The authors find that their C2P method improves model performance compared to not using C2P for GPT-4 on two tasks related to propositional logic queries requiring yes/no answers. An example of a query used for the evaluation is "A, B and C have relations with each other. There is a correlation between A and C, and between B and C. However, A and B are independent from each other. Hypothesis: A directly affects C." for which the answer should be "yes" or "no". The authors also experiment with combining few-shot learning and C2P to find promising results.

**Audience:**

Yes

**Broader Impact Concerns:**

None.

**Claims And Evidence:**

No

**Requested Changes:**

Some proposed adjustments have already been mentioned in the Weaknesses section. Furthermore, I propose the authors make the necessary adjustments to resolve all comments in the Weaknesses section. All aforementioned adjustments are critical to securing my recommendation for acceptance, except for maybe some of the smaller typos mentioned in part 6 of Weaknesses.

**Strengths And Weaknesses:**

## Strengths

- The paper presents "Causal Chain of Prompting" (C2P), a method based on prompt engineering, iterative prompting and the PC algorithm to make LLMs perform better at tasks requiring causal reasoning. It is shown how this approach yields improved results on two tasks that require causal reasoning compared to standard LLMs.

## Weaknesses
I found a total of 6 main points related to weaknesses of the paper, and have listed them with additional comments below:

### 1. The value of the proposed C2P approach is questionable.
1. The proposed method comes with the requirement of processing 9 prompts instead of 1. The method also relies on a model that can accept long inputs (GPT-4 Turbo) which is even more computationally heavy to run. I'm lacking an analysis of the difference in monetary/energy cost of running this approach compared to baseline approaches. This would help show if the C2P method is actually useful.
2. It is unclear if the benefits of the proposed approach is due to the exact prompting, or due to the model being given 9 times more compute power to answer the query. I'm lacking a reasonable baseline corresponding to the same computational cost without relying on the C2P approach to exhaustively investigate this.
3. The performance of the C2P method seems to be evaluated only based on yes/no propositional logic queries (e.g. "A, B and C have relations with each other. There is a correlation between A and C, and between B and C. However, A and B are independent from each other. Hypothesis: A directly affects C."). These queries seem very artificial and I'm not convinced they align with the situations of interest related to LLMs and causality. Neither am I convinced that results on these simple queries would generalise to more interesting situations.
4. The idea of iteratively prompting a LLM to solve some task is hardly novel (see e.g. https://agentgpt.reworkd.ai/ for an example of an implemented solution). Therefore, the proposed C2P approach seems to bring little additional value to the NLP research community.

### 2. There are significant flaws in the method used.
1. Was the C2P approach evaluated based only on comparisons between GPT-4O and GPT-4 Turbo with C2P? Can GPT-4O be fairly compared to GPT-4 Turbo to ascertain the effects of the C2P approach?
2. For the study on the synthetic dataset only 50 samples (out of 200k) from the corr2cause dataset were used. This seems like a very small subset and I'm not convinced that the results presented in table 1 will generalise. Moreover, given the small samples size, I would also like to see the standard deviations for the results in table 1 and a significance analysis. Can the null hypothesis be refuted (i.e. that C2P brings no major advantages compared to standard prompting)?
3. For the study on the "realistic dataset" I'm not convinced that 20 samples is sufficient to prove generalisation of the observed results. Moreover, I have concerns with using a LLM to generate the samples. Did you verify the quality of the samples and assess whether a standard GPT-4 model evaluated on these samples would have an advantage, given the samples having been generated by the same model?
4. For the assessment of the coevolution of supermassive black holes using your proposed framework, it is not clear how this is done or what the objective is. What does it mean that the results provided in Pasquato et al. are replicated? I would appreciate more details on this, together with a motivation and a clear description of the objective with this study.
5. I'm lacking a deeper analysis on whether your C2P method is sensitive to alternative phrasings for the 9 main prompts used, seeing as LMs are well-known to be prompt sensitive. E.g. could even better results have been obtained with a slight rephrasing of your prompts?

### 3. The description of the method is at times unclear, impeding understandability and reproducibility.
1. It is described that "we assess C2P on a proposed synthetic dataset, similar to the one in Jin et al. but more complex and with less information provided on the premises". However, I cannot find any description of how this is achieved. From the paper, it seems as though the discussed dataset is solely based on random sampling from the corr2cause dataset by Jin et al.
2. How many examples were used for the few-shot version of GPT-4 Turbo with C2P? It is described that the number of shorts varies depending on the token limitation of employed LLMs, but I can only see one LLM being used with C2P (GPT-4 Turbo). Later in section 4 for the analysis of the results in table 4 it is described that the few-shot learning process was based on 6 examples - was the same number of samples used for all few-shot model evaluations presented in section 4? It would be good to clarify this.
3. The few-shot C2P baseline should potentially also be referred to as "chain-of-thought prompting" as chain-of-thought prompting is performed in your few-shot setting. Also it is unclear in tables 1 and 3 that "Few-shot learned GPT-4 Turbo" also refers to the C2P based setting, initially it seems as though it is only a "normal GPT-4 Turbo" model. I would recommend adding a "C2P" somewhere in the model name in those tables.
4. How did you generate the 20 natural stories for the realistic scenario evaluation? It would be good if you could add the exact prompts and method to the appendix of the paper, or similarly. Also, it is described that your stories are "more complex" while I cannot find a description or analysis of how this is achieved and/or measured.
5. With respect to the analysis of the results in table 1 it is described how models without C2P that responded correctly had explanations that "often revealed flawed logic" - how was this measured?
6. With respect to the analysis of the results in table 3 it is also described "even when GPTs provide the correct final answer, the reasoning process leading to that answer is still incorrect" - how was this measured?

### 4. The paper fails to address the impact of recent research in their results analysis.
1. Zecevic et al. (2023) whom are cited in the paper, make the claim that current LLMs are unfit to use for causal queries seeing as they are incapable of reliably modelling causal processes. However, this paper seems to make the opposite claim, that LLMs can and should be used for processing causal queries, as long as the correct prompting is used (C2P). I'm lacking a discussion and explanation of how these contradictory results can co-exist.
2. There is also a recent paper by Lu et al. (2024, _Strings from the Library of Babel: Random Sampling as a Strong Baseline for Prompt Optimisation_) that shows how effective prompts might not need to be human readable or task relevant. In the situation of C2P, can we rule out the case that any other iterative prompting process involving 9 steps could result in a performance that is equal to, or even better, than for the C2P method?

### 5. The paper contains many unjustified or imprecise claims.
1. "Causal reasoning is the primary bottleneck that LLMs must overcome to attain human-level intelligence." I find no justifications or references to support this claim.
2. "the inefficiency of LLMs in addressing causal reasoning questions remains their primary bottleneck" I find no justifications or references to support this claim. I would say that the issue with LM hallucinations is also a significant one.
3. "As a result, while they may talk causality, they are not causal", this seems to largely be a repetition of what is stated in the (Zecevic et al. 2023) paper. Regardless, I find this statement imprecise. LLMs based on the Transformer decoder architecture, like GPT, are causal in the way they process input queries which has made them compatible with many interpretability methods based on causal theory, such as causal tracing by Meng et al. (2022, _Locating and editing factual associations in GPT_). So simply stating that they are not causal is incorrect. Maybe something like "LLMs are incapable of modelling causal phenomena as observed in real-world scenarios" would be more precise.
4. "We perform few-shot learning experiments in section 4 with GPT-4 Turbo using our C2P framework and show how integrating C2P during the training or fine-tuning of LLMs can revolutionize existing models." This is a very bold statement, and I cannot see any results involving LLM training to support the statement. The results reported seem to only cover zero-shot and few-short scenarios, involving no weight updates. Moreover, none of the existing results seem to show LMs being "revolutionised".
5. "While current machine learning methods struggle with extracting causal structures and subsequently reasoning causally, this problem is even more intricate in the context of LLMs. The primary reason for this complexity is the inherent structure of LLMs and their reliance on the concept of attention." this claim seems to have been provided without a justification or corresponding reference.
6. The following similar claims can be found in section 5: "... our approach must be integrated into the learning or fine-tuning process of an LLM", "we strongly hypothesize that our approach could similarly benefit the training of language models, particularly in boosting their causal reasoning" and "the fine-tuned or trained LLMs with our approach can achieve highly accurate results". I find no results or motivation to support them, seeing as the results presented in the paper have only focused on few-shot or multiple prompting scenarios (e.g. no weight updates, as would have been involved in fine-tuning and/or training scenarios).

### 6. The writing of the paper is in need of improvements.
There are typos, strange sentences and strange paragraphs in the paper that make it appear half-finished:
1. The second paragraph in the Introduction ("Recently, answering cause and...") feels very long and out of focus. It would be beneficial to move at least a part of this to some related work section. It would also be good if some information could be added here on how your work compares against the related work mentioned - i.e. what is _your_ contribution?
2. The third paragraph in the introduction is hard to follow, and your paper could benefit from some figure that provides an overview of the C2P steps described.
3. Third paragraph in the introduction: "synthethics" should maybe be "synthetic"?
4. Contributions list in the introduction: I do not see the difference between contribution no. 2 and no. 3. Are you not discussing the same results here (section 4)?
5. Section 2 feels very long and I'm not certain all of the information presented there is necessary to keep in the main paper. Maybe you could keep the important parts (e.g. the PC algorithm and brief theory required to understand it) in the main paper and move the rest to the appendix (e.g. do you need to describe do calculus)?
6. Section 2 paragraph 6 ("The be able to perform any level of...") is strange. It starts of discussing the Ladder of Causation, then moves on to discuss causal discovery methods after which LLM implementations are briefly discussed. I would recommend making sure each paragraph in the paper is focused on only one topic or idea, not multiple.
7. Section 2, "PC algorithm:", for points (iv) and (v) in the list the variables $B$, $A$ and $C$ are mentioned without, seemingly, having been described.
8. Section 4.1 contains the following text ".. we set up the following list of LLMs for the experiments on the..." but then fails to directly list the models. I would recommend clarifying this part. Also, I would recommend splitting the paragraph after the models have been introduced, and present the corr2cause dataset in a separate paragraph.
9. "We selected 50 samples at random from the corr2cause dataset... ensuring a balanced distribution with 50 yes and 50 no answers" this does not add up. Did you perhaps select a total of 100 samples at random?
10. It is unclear as to what is meant with the following sentence in section 4 "...the discrepancy between the LLaMa and GPT results [...] stems from the distribution of the provided premise". It would be good to clarify this.
11. It is described how 20 stories are generated for the realistic scenario evaluation, while it seems as though only 10 samples have been evaluated for table 3. Where are the remaining 10 samples?
12. Also related to table 3, it seems as though GPT-4O has been evaluated on 11 samples, while the remaining models have been evaluated on 10?
13. Should "GPT-4.5" be "GPT-3.5" and "GPT-4o" be "GPT-4O" in table 3?
14. I do not understand the point of section 4.4 or what is achieved here.
15. In section 5, I do not understand why the first paragraph is included in the paper. How does it relate to the narrative of the paper and/or the proposed C2P method?
16. There is a missing reference on page 10 for the paragraph that starts with "Scaling Instruction-Finetuned Language Models".
17. There are more typos in the paper than I have managed to mention, I recommend that you run your paper through some spell checker to catch all typos.
18. The flow of the text in the paper could also be improved.

---

> ### Author Response · Authors · 2024-10-09
> **We appreciate the reviewer’s comments and will address them in order. Due to character limitations, we have divided our responses into multiple comments to ensure we address them all. We have incorporated the suggestions into the manuscript and revised it accordingly. The code repository has also been updated with detailed information. We hope that we have adequately addressed all concerns.**
>
> 1-1 C²P has five main subtasks, which run in nine steps. The goal of this paper is to demonstrate that applying this framework during the fine-tuning and training of an LLM can equip it with causal reasoning capabilities, similar to how CoT functions in the current version of GPT. CoT showed that complex tasks can be broken down into subtasks and prompts, allowing an LLM to achieve higher accuracy. Token limitations can occur in CoT as well; however, the goal of the CoT framework was to demonstrate that applying such frameworks in the fine-tuning or training of LLMs, such as GPT, can improve the quality of results. When OpenAI implemented CoT in its training, they reported enhanced response quality. Similarly, for causal reasoning, we do not expect a user to manually perform all nine steps. The simulations of applying the steps one by one are merely proof of concept. For practical use, the few-shot learning process can be employed by a user, allowing LLMs to reason effectively. The key benefit of few-shot learning is that it shows pre-trained models can perform causal reasoning without the need for retraining from scratch or extensive fine-tuning, making the process more cost-effective and compute-efficient. This results in reduced costs and lower infrastructure demands. Therefore, a model fully trained or fine-tuned with C²P on thousands of examples would likely perform even better than one relying on few-shot learning, as it would benefit from training on a larger dataset rather than just the six examples used during few-shot learning—similar to what occurred with CoT. Additionally, we revised the new version of the manuscript to convey the goal of our study better. We have added a paragraph in the discussion section on why including our approach in the training or fine-tuning of LLMs has the potential to revolutionize existing models. We added the results for Llama 3 to show that our framework is generalizable.
>
>
> 1-2 As demonstrated by us and Jian et al. (2023b), current models are unable to perform causal reasoning, even when using CoT. In the introduction section, we reviewed papers that aimed to perform reasoning with external tools. Providing a model with any prompting and nine times more computational power to answer a query does not improve its causal reasoning capabilities. For instance, the current GPT model uses CoT in its responses, particularly when prompted to respond step by step. The results added in the code directory of our study show that LLMs with CoT can involve more than 10 subsequent tasks, yet still fail to reach the correct answer. However, in order to reason effectively, models need to be trained in "how to think," and a specific algorithm, such as the PC algorithm, is required to enable causal reasoning. This limitation is largely due to the design architecture, as highlighted in studies like Kiciman et al. (2023). In the revised version of the manuscript, we have updated both the introduction and discussion sections to clarify the goals and achievements of our study.
>
> 1-3 Causal queries can be divided into two types of questions. The first is a yes-or-no question, known as causal reasoning, and the second is the estimation of effect, known as the Average Treatment Effect (ATE). Our aim was to answer the first type of question, focusing on causal reasoning.  However, users can also ask a quantitative question, such as how much A affects B, which is addressed in studies like Jin et al., 2023a. Such queries require mathematical information in their premise and involve complex computations to derive the ATE, which differs slightly from the primary aims of LLMs. Regarding the significance of the results, the outcomes of few-shot learned C²P on natural stories demonstrate that the framework with LLMs is able to provide better responses, at least compared to using LLMs with CoT and without C²P. As you mentioned, when a query is too complicated, our few-shot learned LLM is unable to answer correctly, but one of the main reasons is the token limitation; with more examples, the results can be improved. We have added multiple examples and their corresponding answers in the code repository to demonstrate the performance of our approach.
>
> 1-4 As you precisely mentioned, the chain of prompting is not new and was not the focus of our study. However, reasoning is one of the main bottlenecks in LLMs that we aimed to address. We show that, with the chain of prompting, these models can learn to think step by step to handle such questions, similar to how humans perform. We used iterative prompting primarily to implement the PC algorithm and demonstrate that it can lead to accurate causal reasoning, which has not yet been addressed in existing studies. The next main goal is to show that through few-shot learning, fine-tuning, or training LLMs with C^2P, these models can respond to new reasoning questions—an important and unresolved challenge for current LLMs, as discussed in the introduction.

---

> > ### Author Response · Authors · 2024-10-11
> > **Response to the second set of comments**
> >
> > 2-1- In the revised version of the manuscript, we have added the results of GPT4 turbo without C2P to better compare the accuracies. To provide even stronger evidence for the generality of our approach, we applied C2P to LLaMA3. The results show that our approach is effective with other LLMs that allow few-shot learning to be applied. We have included the results of applying C2P to LLaMA3.1 in the revised version of the paper. Please note that the previous results for LlaMa were not obtained with zero temperature, which has been corrected in the new version.
> >
> > 2-2 In our study, we compared 120 randomly selected samples, equally distributed between "yes" and "no" answers. To determine the necessary sample size for evaluating the significance of the comparison between the baseline method and models with $C^2P$, we used a sample size formula for comparing two proportions, as presented in Chow SC, Shao J, Wang H, Lokhnygina Y. Sample size calculations in clinical research. Chapman and Hall/CRC; 2017 Aug 15.
> >
> > The formula is : $n = (Z_{α/2}+Z_β)^2 * (p1(1-p1)+p2(1-p2)) / {(p1-p2)^2}$,
> >
> > where $Z_{α/2}$ is the critical value of the Normal distribution at α/2 (e.g. for a confidence level of 95%, α is 0.05 and the critical value is 1.96), Zβ is the critical value of the Normal distribution at β (e.g. for a power of 80%, β is 0.2 and the critical value is 0.84) and p1 and p2 are the expected sample proportions of the two groups.
> > In our case, with confidence of 99, only 117 samples is more than enough to show that the improvement is significant. In the revised version of the manuscript, we mentioned the basis for our sample size and the significance of the results. Since extracting the standard deviation would require running all of our simulations more than 30 times, we have instead provided this statistical test to ensure the significance of the results.
> >
> > 2-3 We have included 30 additional samples in our simulations to ensure that the differences in accuracy are significant, with results confirmed at a 95% confidence level. The main reason for not including more samples in our study is the difficulty of generating and verifying such data. Jian et al. (2023b) discussed how to generate natural stories (we have added more detail in subsection 4.1, and the data themselves are available in the code directory). We have verified them to ensure they are aligned with the facts and excluded those that could not be verified. More details are available in the code repository.
> >
> > 2-4	Thank you for mentioning it. The task is outlined in Figure 1. The goal was to evaluate how C2P performs in real-world scenarios and to determine if C2P can replicate the results. The extracted adjacency matrix, along with the final answer to the question, aligns with the coevolution of supermassive black holes. In the revised version, we have added more details about the task to avoid any confusion for readers and provided additional information on the results and how they align with the referenced study.
> >
> > 2-5 This comment is absolutely valid. Please note that each part of our study serves a specific purpose. The goal of the simulations, where we applied the nine steps of C²P step-by-step, was to demonstrate the expected output and how it leads to causal reasoning. However, our results showed sensitivity to the prompts, as you mentioned. However, in the few-shot learning approach, where examples of expected results for each prompt are provided, this sensitivity is very low. Instead, the results depended on the number of examples. Interestingly, in few-shot learning, although the prompt was lengthy—since all the steps were presented to the LLM in a single prompt—the results remained highly accurate with as few as 10 examples, as reported in Tables table4 and example true story.
> > Additionally, comparing the prompts in Appendices A.2 and A.3, it is clear that the prompts in the few-shot learning section are more abstract and do not need to be as detailed as those used in the step-by-step application of C²P. In summary, while the step-by-step prompting for C²P showed sensitivity at each step, the purpose of this simulation was to prove that the verbalized algorithm can reason causally. On the other hand, the few-shot learning part of the simulation, which is aimed to be used in practice, demonstrated that this framework is not sensitive to the prompts.
> > It is also important to note that the results for natural stories using few-shot learned C2P were highly accurate, even though the few-shot learning examples were synthetic "cause-effect" examples from the COR2CAUSE dataset rather than real-world stories. In the revised version of the manuscript, we have added more details to enhance the transparency of our process and provided additional information on the practical aspects of the study. Furthermore, we have added a new paragraph in the discussion section titled "Practical Insights on Simulations and Results of C²P" to clarify our findings even further.

---

> > > ### Author Response · Authors · 2024-10-11
> > > **Response to the third set of comments**
> > >
> > > 3-1 In Jin et al. (2023b), two datasets are presented: the synthetic data and "Natural Story", which we used both. In the revised version, we added more examples to our experiments to better evaluate our method. Additionally, we have added subsection 4.1 to better explain the data we used in our experiments. Please note that, in corr2cause dataset, the number of hypotheses with "no" answers is significantly higher than those with "yes" answers, leading Jin et al. (2023b) to conclude that only the F1 score can be considered a valid metric. By selecting an equal number of scenarios with "yes" and "no" answers, we aimed to compare the accuracy of our method using different metrics as well, each providing a distinct perspective. We have rewritten the third contribution and also added a paragraph in the discussion section, "Why does integrating C2P during the training or fine-tuning of LLMs have the potential to revolutionize existing models?",  to give a better understanding of the presented contribution. We have revised the abstract and introduction to address your comment better as well.
> > >
> > > 3-2 In the revised version, we have added the mentioned materials in the "Few-shot Learning Agenda" subsection. Also, an example of a prompt for few-shot learning is available in the code directory of our study, so to use it, simply copying and pasting the prompt will suffice. We have also included the few-shot learning results for LlaMA using the same examples and added the results to the existing tables. Please note that the number of examples is limited by the token limit of the model, and increasing the number of examples improves the accuracy of the answers. We mentioned that we use 10 examples in the manuscript to avoid any confusion. This is one of the main reasons we suggest that using C2P in the fine-tuning of a model can lead to even higher accuracy, as hundreds of examples can be used in that process.
> > >
> > > 3-3 Thank you for pointing that out. When responding to reasoning questions, we instructed the models to think step by step to ensure the Chain of Thought (CoT) mechanism was activated. In the revised version, we have mentioned this at the end of the "Experimental Setup" subsection. Additionally, we included C²P alongside the few-shot learning method to avoid any confusion.
> > >
> > > 3-4 We have added the natural stories, along with an example prompt to generate them, in the code repository of the study. In the revised version, we omitted the term "more complex," as it would require quantitative metrics to compare with the data in Jian et al., 2023b. The complexity we originally referred to was based on the diverse fields from which the natural stories were generated.
> > >
> > > 3-5 As you pointed out, it would be insightful to provide an example of how LLMs without C^2P exhibit flawed logic, even when prompted to reason step by step (Chain of Thought). We have added an "Example of Comparison in Responding to Causal Queries" in the appendix A.2, demonstrating how a model can reach the correct final answer while using flawed underlying logic. Additionally, multiple examples of LLM responses are available in the code repository, where hallucinatory behavior and reasoning can be observed.
> > >
> > > 3-6 This is similar to what we addressed in your previous comment. The evaluation process is straightforward—we focus on the arguments presented by the LLM. Please note that when calculating the accuracy metrics, we consider only the final answer, not the reasoning process, to maintain consistency with existing studies. Highlighting instances of flawed logic, despite a correct final answer, helps clarify why we suggest that the models may be acting randomly, beyond their performance as reflected in the reported metrics.

---

> > > > ### Author Response · Authors · 2024-10-11
> > > > **Response to the fourth set of comments**
> > > >
> > > > 4-1 We believe that the claim made by Zecevic et al. (2023), stating that current LLMs are unfit for causal queries is entirely valid, as we have mentioned in both the introduction and discussion sections. This limitation stems from their inherent structure, which focuses on pattern recognition without the capability to discern causal relationships, as indicated by prior research (Kıcıman et al., 2023).
> > > > In developing LLMs, data for multiple tasks are incorporated during training to ensure that the models can handle those tasks effectively. Our goal is to demonstrate that by verbalizing an algorithm similar to the PC algorithm and using it step by step in few-shot learning, training, or fine-tuning processes, these models can extract the adjacency matrix of causal relations based on the associations in the given premise. Then, LLMs such as GPT-4 can determine cause and effect based on this adjacency matrix. As Judea Pearl has shown, statistics alone cannot inherently perform causal inference; however, statistical tools can facilitate causal reasoning. For instance, in Spirtes et al., 2001, the PC algorithm was proposed to extract a causal graph based on associations in a causal system.
> > > > We have revised both the introduction and discussion sections to clarify the goals and approach of our study. Additionally, we included more detail on the ladder of causality proposed by Judea Pearl to help readers better understand the perspective of our study. We have also added a new paragraph, "Why integrating C^2P during the training or fine-tuning of LLMs has the potential to revolutionize existing models?" in the discussion section to better address your comment.
> > > >
> > > >
> > > > 4-2 This is correct, and we observe it in our results. Other prompts capable of executing the algorithm we used can be applied interchangeably, which supports the idea presented in the reference you mentioned. This is especially evident in the few-shot learning process, where providing more examples has a significant impact; even more abstract prompts can achieve similar or better results. As long as a prompt clearly asks LLMs what the result should be and what information is given to reach that result (i.e., it follows the steps defined in the algorithm), the prompt can be interchangeable. We have added the reference you mentioned to the same paragraph in our study to provide readers with a clearer perspective on the concept we are discussing.

---

> ### Author Response · Authors · 2024-10-11
> **Response to the fifth set of comments**
>
> 5-1 Current AI mostly focuses on identifying dependencies among variables and discovering patterns. LLMs are no exception; both the encoder and decoder structures in LLMs explore patterns with an emphasis on the concept of attention. Another example is the autoregressive structure used in some LLMs. Judea Pearl's ladder of causality illustrates how human intelligence operates and outlines the steps to achieving human-level intelligence.
> The first step is essentially "association," which is limited to answering questions like "What is...?" or "How would seeing X change our belief in Y?" The second and third steps pertain to action, imagination, and retrospection. By progressing to these steps, one can engage in causal reasoning, addressing questions such as "What if...?", "What if I do X?", "Why?", and "Was it X that caused Y?" Consequently, a model must possess causal reasoning capabilities to respond effectively to reasoning questions.
> In the new version, we have revised the introduction of the manuscript to better address this claim. We have added more detail on the ladder of causality to provide further support for our arguments.
>
> 5-2 Studies such as Xu et al. (2023), Nezhurina et al. (2024), Jin et al. (2023), and Romanou et al. (2023) highlight reasoning, as a significant challenge or bottleneck for LLMs, requiring further research and improvement. However, since there is no definitive metric to determine whether reasoning or hallucination is more important, in the revised version of the paper, we changed the phrase "the main bottleneck" to "one of the main bottlenecks." Please note that these two topics are not entirely distinct. For example, Kalai et al. (2023) discuss how LLMs produce hallucinations and lack true causal reasoning, even with access to vast amounts of data. The authors argue that while LLMs can mimic causal language, they fail to engage in genuine causal reasoning.
>
> 5-3 As the respected reviewer mentioned, the structure of LLMs can utilize bidirectional or unidirectional data. The term "causal" in this context refers to the fact that only past words are used to generate new words. While this is entirely accurate, it is distinct from the concepts of causality and reasoning discussed in our study. We believe that Zecevic et al. (2023) primarily found that the reasoning provided by LLMs merely mimics the data present during the training phase and does not reflect an inherent capacity for causal thinking. This is the main reason we use the term "causal reasoning" rather than just "causal." In the new version of the manuscript, we revised any sentences that might cause confusion between these two terms.
>
> 5-4 To address the comment, we have revised the mentioned paragraph in the new version of the manuscript as follows:
> We perform few-shot learning experiments in Section 4 with GPT-4 Turbo and LLaMa 3.1 using our C^2P framework to propose that integrating C^2P during the training or fine-tuning of LLMs has the potential to revolutionize existing models in responding to reasoning queries.
> Additionally, we have added the section titled "Why does integrating C^2P during the training of LLMs have the potential to revolutionize existing models?" in the discussion section. Given that reasoning is one of the most significant bottlenecks in the field and that no methods have been proposed in the literature for enhancing the causal reasoning of LLMs, along with the improvement in reasoning, we observed, we believe that C^P represents a significant advancement in this area.
>
> 5-5 In the revised version of the manuscript, following the sentence you mentioned, we have provided some foundational support for our claim.  We revised this paragraph and explained that the concept of attention is akin to the concept of "association," as described by Pearl in his ladder of causality. We believe this addition enhances the clarity of our discussion.
>
> 5-6 Our study has two main goals. First, we aim to demonstrate that by employing the proposed algorithm with nine steps (five subtasks), LLMs can perform reasoning. Second, we utilize this method in few-shot learning to illustrate its practical utility. By using few-shot prompts, one can pose reasoning questions, and we show that few-shot learning with as few as ten examples can enhance a model's reasoning capabilities. Consequently, we expect that fine-tuning or training a model with hundreds of examples using our framework can achieve even higher accuracy in reasoning. As described in studies such as Chung et al. (2024), Kaplan et al. (2020), and Brown et al. (2020), among others, the scaling law supports the same concept we have discussed.
> In the new version of the paper, we have revised the first sentence you mentioned, omitted the second sentence, and updated the third one based on the explanation provided in response to this comment.

---

> > ### Author Response · Authors · 2024-10-11
> > **Response to the sixth set of comments**
> >
> > 1-  We revised this paragraph and tried to present it more organized by dividing the studies into two main groups: the study that evaluates the ability of LLMs to reason and the proposed methods.
> > 2- In addition to figure 1, we have added figure 2, but in section 2 to better illustrate the 5 main steps. We tried to better explain our contributions in paragraph 3 of the introduction.
> > 3-It is fixed in the new version.
> > 4- We have revised the contributions to distinguish the second and third ones better.
> > 5-We have revised Section 2, reorganizing and rewriting the material. A new figure has been added, and the referenced paragraph has been moved to the appendix.
> > 6- We added a specific paragraph in section 2 on the ladder of causality.
> > 7-This has been corrected in the new version.
> > 8- we have added a new subsection, 4.1, to better describe the dataset. We also revised the paragraph you mentioned.
> > 9- We revised this in the new version, selecting 120 samples, with 60 samples for "yes" answers and 60 for "no."
> > 10,11,12 and 13- we revised the manuscript based on the mentioned comments.
> > 14- The goal is to apply our framework to real-world data. We have added more details about this data in Subsection 4.1 to better clarify the purpose of the experiment.
> > 15-This is a very important technical discussion. To effectively learn a causal system, the system's identifiability is crucial—essentially determining whether the provided information is sufficient to learn the causal graph and subsequently engage in causal reasoning. Additionally, it is important to understand what can be extracted from a PDAG  and what additional information is essential to fully reason within a given premise.
> > 16,17 and 18- we revised the manuscript based on the mentioned comments.
> >
> >
> > Thank you for mentioning all the 18 points. In the revised version, we addressed all of these issues and tried to revise the manuscript's overall writing. Please note that additional revisions have been made based on feedback from other reviewers. We have added Figure 2 as an illustrative example of our framework. A figure was added to better illustrate the framework. The details on do-calculus are moved to the appendix

---

> > > ### Comment · Reviewer_54PX · 2024-10-14
> > > **Response to author comments**
> > >
> > > Thank you for the exhaustive answers to my concerns. You can find my response corresponding to each point of weakness below.
> > >
> > > ## 1. The value of the proposed C2P approach is questionable.
> > >
> > > > a model fully trained or fine-tuned with C²P on thousands of examples would likely perform even better than one relying on few-shot learning, as it would benefit from training on a larger dataset rather than just the six examples used during few-shot learning—similar to what occurred with CoT
> > >
> > > Still, there are no experimental results in your paper to support this claim. I find no experiments in your paper that involved the training of a LM with C2P. I see that reviewer 3jUT shares the same concern.
> > >
> > > > Providing a model with any prompting and nine times more computational power to answer a query does not improve its causal reasoning capabilities. For instance, the current GPT model uses CoT in its responses, particularly when prompted to respond step by step. The results added in the code directory of our study show that LLMs with CoT can involve more than 10 subsequent tasks, yet still fail to reach the correct answer.
> > >
> > > It is great that you have experiments on this! However, I can't see that you have added any mention of this in your paper? I assume that results from this would be added to your paper, to clearly show that C2P actually brings more value compared to CoT, and at a similar computational cost?
> > >
> > >
> > > However, my main concern related to the value of your method remains. C2P has so far only been tested on very synthetic samples (e.g. "Premise: Let’s consider three factors in a marketing context: advertising spend (Z), brand awareness (Y), and sales revenue (X). There is a correlation between advertising spend and brand awareness, and between advertising spend and sales revenue. Additionally, there is a correlation between brand awareness and sales revenue.") and I do not find the results on those samples sufficient to support the claim that C2P equips LLMs with causal reasoning capabilities, as stated in your paper. The C2P method used seems very forced and my fear is that you get good results on your synthetic datasets simply because the pattern of those samples aligns with the C2P method.
> > >
> > > ## 2. There are significant flaws in the method used.
> > >
> > > Thank you for clarifying these things! Thus, I generally consider this point of weakness as resolved.
> > >
> > > However, I'm still confused about the experiment described in section 4.5. For example in figure 4 the output is "A: Yes". What question does this answer? Also, it is still unclear from the paper as to whether figure 4 shows the conceptual framework for the task or the actual results of C2P.
> > >
> > > ## 3. The description of the method is at times unclear, impeding understandability and reproducibility.
> > >
> > > Thank you for adding the details on this to the paper.
> > >
> > > ## 4. The paper fails to address the impact of recent research in their results analysis.
> > >
> > > Thank you for adding the details on this.
> > >
> > > ## 5. The paper contains many unjustified or imprecise claims.
> > >
> > > Thank you for addressing my comments on this. I believe most of my concerns for this part have been resolved.
> > >
> > > However, I still disagree that you have found sufficient evidence to state that "integrating C2P during the training or fine-tuning of LLMs have the potential to revolutionize existing models". See my comments in part 1.
> > >
> > > ## 6. The writing of the paper is in need of improvements.
> > >
> > > Thank you for making these edits.

---

> > > > ### Author Response · Authors · 2024-10-15
> > > > **We aimed to address your concerns in the last revision of the manuscript.**
> > > >
> > > > Thank you so much for your insightful comments. Your previous precise, valuable, and detailed comments, along with the new feedback, have significantly helped clarify the existing challenges and greatly improve the manuscript.
> > > >
> > > > 1-1 We have updated the manuscript based on your and the other reviewer’s comments, omitting this from the contribution section since we are unable to perform an experiment on it. Additionally, we removed the paragraph titled “Why Does Integrating C2P During the Training or Fine-Tuning of LLMs Have the Potential to Revolutionize Existing Models” from the “Discussion, Limitations, and Future Work” section. Instead, we expanded the future work paragraphs, adding more references for similar studies and providing further details on fine tuning discussion. We believe this will help readers better understand the perspective of our study. As a result, we or someone else can design the experiment and perform the fine-tuning procedure to evaluate the hypothesis. If you believe any of this is unnecessary or potentially misleading, we can make further revisions as needed.
> > > >
> > > >
> > > > 1-2 Based on your comment, we have added an entirely new paragraph in the “Practical Insights on Simulations and Results of C2P” subsection. We compared our framework with CoT and mentioned that while the CoT aims to break reasoning tasks into multiple simpler tasks, it often fails to do so. This challenge is also highlighted in Wei et al. (2022b), where CoT is introduced and its logical inconsistencies and poor step alignment are identified as key limitations of the approach.
> > > >
> > > >
> > > > 1-3 The main goal of designing our 3 levels of experiments was to address your concern, which is also a concern of ours. After applying our framework to the synthetic data, we aimed to evaluate it on the “natural stories”; however, they also remain partially synthetic. The goal of our experiment on the black hole example was to test it in a fully real-world scenario. It is important to note that many real-world examples do not provide similar information as our examples. In Ceraolo et al. (2024), the authors trained a model to evaluate a given premise first to determine whether the provided information constitutes a causal question (as we previously mentioned, this engine can be used prior to our framework in training LLMs). We attempted to evaluate the reasoning challenge from multiple aspects; however, there are scenarios that are more complex than our framework might be able to handle. Based on our exhaustive literature review, no framework has yet been presented to tackle reasoning challenges in LLMs. Our proposed method can serve as a starting point to address this bottleneck in these models. We do not expect our method to completely solve the reasoning problem, but similar to the development of any other method that progresses step by step, our approach can be the first step forward in addressing this significant inefficiency. We hope that our explanation addresses your concern.
> > > >
> > > >
> > > > 2- In the revised version, we updated subsection 4.1 and added the exact premise and causal hypothesis. We also revised subsection 4.5 to clarify that Fig. 4 presents the actual results of our framework using GPT. We also mentioned that other questions can be answered by the extracted PDAG. We hope that we have addressed your comment clearly in the new version.
> > > >
> > > >
> > > > 5- As we addressed this comment in Part 1, we have omitted it from the contribution section and the “Why Does Integrating C2P During the Training or Fine-Tuning of LLMs Have the Potential to Revolutionize Existing Models” subsection. In the future work paragraph, we have added more details in the fine tuning LLMs with our framework and mentioned similar studies. As mentioned, if you believe that even this paragraph is unnecessary, we can remove it.

---

> > > > ### Author Response · Authors · 2024-10-19
> > > > **Further revisions based on your comments and those of the other reviewers**
> > > >
> > > > We have updated the manuscript and revised it in different sections. We specifically revised the structure of the "Discussion, Challenges, and Future Work" section in the updated version to incorporate your feedback. Your comments were included as challenges and future steps toward achieving reasoning in complex scenarios.
> > > >
> > > > Regarding the statement, "As a result, we believe that the impact of C2P on LLMs with causal reasoning capabilities is similar to the transformative impact of ‘Chain-of-Thought’ (Wei et al., 2022b) on LLMs, as discussed by (Chung et al., 2024)" , other reviews and you mentioned, we have completely revised it. In the revised version, after addressing the challenges, we state that "By overcoming these challenges, the integration of C2P with LLMs can provide these models with causal reasoning capabilities, similar to the transformative impact of "Chain-of-Thought" (Wei et al., 2022b), as highlighted by Chung et al. (2024)"
> > > >
> > > > We hope that the revisions to the manuscript have clarified the content and addressed the challenges, making it easier for other researchers to build upon our work. Your feedback, along with that of the other reviewers, has significantly enhanced the quality of the study.

---

### Review · Reviewer_4X51 · 2024-09-20

**Summary Of Contributions:**

This paper introduces Causal Chain of Prompting (C2P), a framework for conducting causal reasoning within LLMs in a chain-of-thought-like manner, without relying on external tools. The C2P framework uses 5 reasoning steps, which include: (i) extracting the random variables in the data; (ii) extracting all the cause and effect relations (along with the conditional and unconditional relations) among the random variables; (iii) creating an initial adjacency matrix; (iv) extracting the causal partially directed acyclic graph; and (v) prompting for cause-and-effect questions and hypotheses.

Experiments on synthetic and real-world domains demonstrate that, when applied to leading LLMs, the C2P approach outperforms the baselines on causal reasoning tasks.

**Audience:**

Yes

**Broader Impact Concerns:**

No broader impact concerns come to mind.

**Claims And Evidence:**

No

**Requested Changes:**

1. **Critical**: clarifying various aspects of the experimental setup and evaluation tasks, metrics, etc. (see Weaknesses 1-3 above).

2. **Critical**: refactoring and shortening the introduction section to focus more on the exact contributions of the paper (see Weakness 4 above).

**Strengths And Weaknesses:**

# Strengths

1. The question of how we can conduct causal reasoning with LLMs is an important open research question in the field.

2. The paper includes a comprehensive review of causal learning and reasoning, which can help readers who are directly familiar with the field to get up to speed with the paper's approach.

# Weaknesses

Despite the strengths, the paper has some serious weaknesses:

1. The primary weakness relates to the clarity of the experimental setup and the evaluation numbers. To that end, Section 4.1 explains the experimental setup, but many important aspects are unfortunately still left very much unclear. For example:
    - The paper simply mentions that the first synthetic dataset is similar to the experimental setup of (Jin et al., 2023b) for experiments on the CORR2CAUSE dataset, and that the paper only includes "powerful LLMs" because Jin et al. (2023b) already explored other LLMs. This description is not particularly comprehensive, and leaves the readers who are not familiar with the work of Jin et al. (2023b) confused. What exactly is included in the CORR2CAUSE dataset? Is there an example format of the task? What is the task exactly? What is the evaluation metric? I am not hoping for a comprehensive description, but some further context about the empirical setup is very much needed here to properly evaluate the paper.
    - Another under-explained empirical setup is "... we assess C2P on a proposed synthetic dataset, similar to the one in (Jin et al., 2023a), but more complex and with less information on the premises." This is unfortunately also not descriptive at all: What does the premise look like exactly? How does it differ from (Jin et al., 2023a)? Why exactly is it more complex? Would the result be directly comparable to theirs?
    - Section 4.1 mentions that there is a balanced distribution with 50 Yes and 50 No questions. What does the task look like, and how are the simple Yes / No answers obtained? Some more details here, particularly regarding the CORR2CAUSE dataset, would be much appreciated.
    - It seems like, to obtain the natural stories in Section 4.1, the authors generate stories with GPT4. What are the prompts and what do the stories look like? Is it directly comparable to prior work? To what extent do the stories incorporate causal relationships? These aspects are still left unclear.
    - What exactly are the first, second, etc. sub-tasks in Table 2?

2. Another unclear aspect is in section 4.4 (evaluation of the C2P framework on the (co)evolution of supermassive black holes and their host galaxies). Figure 3 only shows the resulting identified causal graph. Is there a quantitative assessment as well on how well the model performs? For instance to evaluate the resulting correctness in the causal graph, and also the final output. It seems like the evaluation here is only qualitative in nature. Furthermore, how well do the baseline approaches do here? This aspect is also still left unclear.

3. It is still unclear to me why the C2P approach is only applied to GPT-4 Turbo in Table 1, and not to the LLaMa-3 model as well. It seems to me like both of these models are autoregressive LLMs, so the proposed technique should be able to be applied to both, and not just to GPT-4?  If the approach can improve performance on both LLaMa-3 and GPT-4, this would present stronger evidence to the generality of the approach.

4. The introduction (one of the most important sections of any paper) is written more like a related work section, and is quite belaboured. It would be great to make it more succinct, and focus more on the exact contributions of the paper, rather than reviewing prior work in too much detail.

---

> ### Author Response · Authors · 2024-10-08
> **We thank the reviewer for their comments, which we address in order. Due to the character limitation, we briefly explain our idea for your feedback and detailed the revisions made to the manuscript, in response to your comments. We hope that we have adequately addressed all of them.**
>
> 1-1  We added a new subsection (4.1 Datasets) to the new version of the manuscript, which provides a brief introduction to the datasets and explains how they were generated. We have provided an example of each and also mentioned that more examples are in the project code directory. Also, we have added sample responses for LLMs with and without $C^2P$ to give a better understanding on how they perform. Additionally, the term "powerful" is qualitative rather than quantitative and omitted. In the revised manuscript, we have updated that section of the paragraph accordingly.
>
> 1-2	 The problem mentioned as under explainbitly is addressed in the new version of the manuscript. The Corr2cause dataset did not share its data in the natural story section, only explaining how it was generated. We removed the phrase "... we assess C2P on a proposed ..." and instead provided an explanation of the datasets in subsection 4.1. The original phrasing was used because we asked GPT to generate complex scenarios based on the provided information. Additionally, More details are added in the code repository. The results are comparable, but there are some important considerations. The goal of their study was to provide a benchmark dataset, and the responses in their data are predominantly "No." This is because they aimed to explore all possible scenarios, many of which yield a "No" result. However, a limitation of their study is that the metrics (F1, recall, precision, and accuracy) do not transparently highlight the differences between models, as they noted. To address this, we randomly selected samples from their dataset while ensuring that the "Yes" and "No" responses were balanced. This approach offers a clearer understanding of the current models' performance based on the reported metrics. For instance, as shown in Table 1, LLaMA models tend to respond "Yes," while GPT models are more inclined to respond "No", which is not discovered in their studies. The exact responses from some of our simulations have been added to the code repository of our study to provide a better understanding of their performances.
>
> 1-3	 The examples provided in subsection 4.1 of the revised manuscript address the comment. We used the sample size formula to compare two proportions and showed that with 117 examples, the improvement was significant, with 99% confidence. We have added 20 more examples and discussed that this ensures the significance of the improvement in reasoning. In the revised version, this is discussed after Table 1. Please note that the results in the previous version of the paper for LlaMa were without zero temperature and we updated them in the new version with zero temperature.
>
> 1-4	   Jin et al. (2023b) described the process of generating the data but did not make the data publicly available, only providing an example. In the code repository for our study, we have included an example prompt for generating such data, along with the relevant code details. We believe the data would be similar. The COR2CAUSE study has not reported results for using LLMs to respond to such queries. All of this is discussed in subsection 4.1 of the revised manuscript.
>
> 1-5	 The five subtasks are outlined in both the Introduction and Method sections. These subtasks are applied with nine prompts. In the new version of the manuscript, we also attempted to indicate which subtask corresponds to each prompt in subsection 3.1.
>
> 2- Proceeding with $C^2P$ involves five subtasks, with the expected results being quantitative. The study by Pasquato et al. (2023) confirmed the hypothesis that central density affects black hole mass by extracting its PDAG, which we have replicated through subtasks 1-4 and then answered the hypothesis with subtask 5. The primary difference between our study and theirs is that we have verbalized their data to demonstrate that $C^2P$ can achieve the same conclusions autonomously as a rational human through logical reasoning. More discussion on these is provided in the new version of the manuscript (after Figure 4).  It is important to note that in some cases where existing LLMs are asked a reasoning question, the final answer may be correct, yet the underlying logic is completely flawed and hallucinatory (An example is provided in the appendix).
>
> 3- Thank you for mentioning it. We have included results for LlaMa-3.1 in the revised manuscript to highlight the generalizability of our approach. Additionally, we have included results from few-shot learning using $C^2P$ on LlaMa-3.1. These results confirm that our framework functions effectively on these LLMs. It is worth noting that the operational mechanics of LLMs, such as the autoregressive process, do not hinder the effectiveness of $C^2P$, as long as the models can execute chain-of-prompting reasoning.
>
> 4- We have extensively revised the abstract and introduction in the new version based on your comments and feedback from other reviewers.

---

> > ### Comment · Reviewer_4X51 · 2024-10-18
> > **Thank You For Your Response**
> >
> > I would like to thank the authors for their thorough authors' response and revision, which improves upon the original submission. I have also carefully read the other reviews, and the authors' response to those reviews. Regarding the authors' response to my comments:
> >
> > 1-1 Thank you for adding more explanation about the datasets. I find the current version to be much clearer, and I am able to follow the empirical results much better.
> >
> > 1-2 This makes sense, thank you for the explanation. Please also mention these points in subsequent versions of the manuscript (i.e. that some of the Corr2Cause dataset, in particular the natural stories part, was not open-sourced, so it needs to be regenerated with the same methodology using GPT-4).
> >
> > 1-3 Thank you for adding the examples.
> >
> > 1-4 This makes sense, thank you for clarifying.
> >
> > 1-5 This also makes sense, thank you for clarifying.
> >
> > 2 Thank you for adding more details and explanation about this evaluation; it is much clearer now.
> >
> > 3 Thank you for adding the LLaMa results. It is encouraging that the improvements in causal reasoning are also observed by applying C2P on top of LLaMa in addition to GPT-4, which demonstrates the generalisability of the approach.
> >
> > 4 Thank you for the revised abstract and intro. It is better than the original version. One suggestion is to revise the introduction section a little further, particularly regarding the second paragraph in page 2 (the paragraph that spans the longest length), which still reads a bit like related work to me. My suggestion is to focus on a few key work highlighting the limitations of causal reasoning in LLMs, and leave the rest for the related work section. This will make the introduction section more succinct and coherent.

---

> > ### Comment · Reviewer_4X51 · 2024-10-18
> > **Further Thoughts**
> >
> > Having read the other reviewers and the authors' response to them, I have some further thoughts regarding the work.
> >
> > It is encouraging that the paper demonstrates strong improvements on the Corr2Cause, Natural Stories, and Supermassive Black Hole datasets. However, I share Reviewer 54PX's ongoing concern regarding the **synthetic** nature of the Corr2Cause and Natural Stories datasets, which might limit the broader applicability of the proposed approach. While I acknowledge that the Natural Stories dataset is more realistic than the Corr2Cause dataset, it is still the case that this dataset was **generated by GPT-4**, which means that it is still synthetic data. Moreover, the template of this dataset is designed such that it is very much amenable to causal reasoning, which might overstate the impact of the proposed method on other, more realistic datasets & LLM use cases.
> >
> > For example, page 7 provides an example of the Natural Stories dataset, which goes as follows: "Premise: Let’s consider three factors: eating junk food, obesity, and watching television. There is a correlation between eating junk food and obesity, and between watching television and obesity. However, eating junk food and watching television are independent from each other. Hypothesis: Eating junk food directly affects obesity."
> >
> > This example still seems very synthetic to me because (i) the variables are clearly given to the model (i.e. there are 3 variables here: Eating junk food, obesity, and watching TV, and **the 3 variables are immediately given to the model**); (ii) the paper only goes up to 6 variables (see Table 2); and (iii) there is not much ambiguity in the dataset regarding what the variables, hypothesis, and premise are (more on this below), which makes causal reasoning much easier.
> >
> > In practice, I am not sure whether these 3 assumptions hold in real-world use-cases of LLMs. In practice, in most real-world LLM use cases, (i) it would not even be clear what the variables are (i.e. the model would have to infer what the variables are, which is a noisy process during which the model can identify the wrong variables, or only identify a subset of the variables, which would mess up step 1 in C2P, and by extension, all the other steps going forward from there can go wrong); (ii) the hypothesis and premise might be latent and not clearly stated (e.g. the model needs to infer whether or not person A believes in a particular hypothesis, and identify what the premises were, all based on limited information from the text, which is again a noisy process); and (iii) there may be more than 6 variables involved, which can hurt model performance by a lot (in Table 2, going up to 6 variables already reduces the accuracy of the fourth & fifth steps; in practice, even step 1 which identifies all the variables would **not** have 100% accuracy in the face of noisy real-world data, which might mess up all subsequent steps and reduce model performance by a lot).
> >
> > Related to this point, page 13 mentions that "As a result, we believe that the impact of C2P on LLMs with causal reasoning capabilities is similar to the transformative impact of ‘Chain-of-Thought’ (Wei et al., 2022b) on LLMs, as discussed by (Chung et al., 2024)". I am still skeptical regarding this point, because CoT prompting is a lot more generally applicable than C2P; after all, CoT involves mostly appending the model's intermediate output into the context, whereas the C2P approach makes many assumptions that may not hold in real-world LLM use cases (i.e. that it is clear what the causal variables are, that we don't go beyond 6 causal variables, that the premise & hypothesis are explicitly stated, etc.).
> >
> > I fully understand that one paper cannot possibly cover everything, and that evaluating causal reasoning in real-world LLM use cases on non-synthetic data is still difficult to do (let alone making a lot of progress there, which is very hard without good evaluation). However, I expect there to be more discussion around these limitations, and ideally with a non-synthetic or more realistic evaluation that shows how C2P performs in cases where the variables or premise or hypothesis are not explicitly stated and need to be inferred by the model, which I imagine would be a very noisy process that can non-trivially reduce the performance of the approach under more realistic conditions.

---

> ### Author Response · Authors · 2024-10-19
>
> Thank you for your comments. We have addressed the remaining concerns in this version. Specifically, we revised the "Discussion, Challenges, and Future Work" section and its structure in the updated version to incorporate your feedback. We included your comments as challenges and future steps for achieving reasoning in complex scenarios.
>
> Regarding the statement, "As a result, we believe that the impact of C2P on LLMs with causal reasoning capabilities is similar to the transformative impact of ‘Chain-of-Thought’ (Wei et al., 2022b) on LLMs, as discussed by (Chung et al., 2024)," we have completely revised the paragraph. In the revised version, we address the challenges and state: "By overcoming these challenges, the integration of C2P with LLMs can provide these models with causal reasoning capabilities, similar to the transformative impact of "Chain-of-Thought" (Wei et al., 2022b), as highlighted by Chung et al. (2024)"
>
> We hope the revisions to the manuscript provide clearer materials and effectively address the challenges, allowing other researchers to build on our work. We believe that your comments, along with those of the other reviewers, have been invaluable in improving the study.

---

### Decision · Action_Editor_J9cH · 2024-10-28

**Recommendation:** Reject

**Comment:**

* The method is given only a little over a page of space in the main paper, and its description does not cover which 9 prompts are actually used. It it is well-known in LLM research that the formulation of the prompt itself can change results dramatically. This is not investigated at all in this submission.

* Minor comment: There are several other papers investigating causal reasoning with LLMs that are not discussed here, see e.g., [1, 2, 3, 4].

[1] Kıcıman, Emre, et al. "Causal reasoning and large language models: Opening a new frontier for causality." arXiv preprint arXiv:2305.00050 (2023).
[2] Liu, Xiao, et al. "Are llms capable of data-based statistical and causal reasoning? benchmarking advanced quantitative reasoning with data." arXiv preprint arXiv:2402.17644 (2024).
[3] Hobbhahn, Marius, Tom Lieberum, and David Seiler. "Investigating causal understanding in LLMs." NeurIPS ML Safety Workshop. 2022.
[4] Bao, Guangsheng, et al. "Llms with chain-of-thought are non-causal reasoners." arXiv preprint arXiv:2402.16048 (2024).

**Audience:**

* There is an audience for work on LLMs and causality within the TMLR community and we should be careful publishing poorly supported papers for that audience.

**Claims And Evidence:**

All three reviewers had concerns with the paper not supporting its claims, even after multiple responses and revisions by the authors. The weakest supported claims made in the abstract and introduction are as follows:
* "[The method] can be seamlessly integrated into the training or fine-tuning of LLMs." ... "The improvement observed in both fewshot learned GPT-4 Turbo and LLaMA 3.1 provides evidence of the generalizability of C2P, highlighting its potential to be incorporated into the training or fine-tuning of new LLMs to enhance their reasoning capabilities"
  * This has not been demonstrated empirically or theoretically
* "We first demonstrate that reasoning accuracy improved by over 30.7% and 25.9% ..."
  * This is on a task on which a supposedly random baseline (with balanced data) achieves 70% accuracy. (see more below)
* "In this paper, we propose the first reasoning framework for LLMs, called the Causal Chain of Prompting (C2P), designed to enhance reasoning skills by climbing the causality ladder to address reasoning questions."
  * Only interventions (not counterfactuals) are considered. So the "climbing" is limited to one step above methods that do not claim to reason about interventions. Moreover, the framework comes with no guarantees that interventional parameters will be identified.
* "Through extensive experiments with our framework, we demonstrate a significant improvement of LLMs in causal reasoning in various benchmarks. Additionally, we examine the performance of the C2P framework on more complex and real-world scenarios in various domains"
  * Several reviewers took issue with this description as the bulk of the results are limited to two synthetic datasets of similar structure. Moreover, the number of examples from those data sets are very small.
  * On CORR2CAUSE, the two random baselines (uniform and [label-]proportional) greatly exceed the label rate in accuracy (which is 50%). This was questioned by a reviewer but it was never explained. Since the authors clearly state that the data set is balanced, a) the two random baselines should be identical, and b) if the results are to be believed, there is a great degree of variance. (The probability of seeing more than 70 heads in 100 tosses of a fair coin is less than 1 in 10000). The authors claimed to have performed a significance test to establish the superiority of their method over LLM baselines, but it would seem that the random baseline would pass that test as well, hinting that there is an error made somewhere.
* The third contribution stated in the introduction is not a claim that can be tested.

Additionally, despite reviewers pointing out that this claim is not supported, the authors have left a comment stating "This highlights how integrating C2P during the training and fine-tuning of LLMs can revolutionize existing models."

In short, this paper makes several unsupported or weakly supported claims, including about large and lasting impact on the field of LLMs as a whole. Despite several revisions, this aspect of the manuscript has not been improved. It would be appropriate to either significantly scale up experiments or to carefully rewrite the framing of this paper as a small-scale investigation into a particular series of prompts aimed at extracting causal understanding from LLMs, as part of chain-of-thought reasoning.

**Resubmission Of Major Revision:**

The authors may consider submitting a major revision at a later time.